

# Calcification and distribution of extant coccolithophores across the Drake Passage during late austral summer 2016

Mariem Saavedra-Pellitero[1,2], Karl-Heinz Baumann[1], Miguel Ángel Fuertes[3], Hartmut Schulz[4], Yann Marcon[1,5], Nele Manon Vollmar[1], José-Abel Flores[3], Frank Lamy[6]

[1] Department of Geosciences, University of Bremen, PO Box 33 04 40, 28334 Bremen, Germany.
[2] School of Geography, Earth and Environmental Sciences, University of Birmingham, UK.
[3] Departamento de Geología, Universidad de Salamanca, 37008 Salamanca, Spain.
[4] Mathematisch-Naturwissenschaftliche Fakultät, University of Tubingen, Hölderlinstr. 12, 72074 Tübingen, Germany.
[5] MARUM, Center for Marine Environmental Sciences, University of Bremen, 28359 Bremen, Germany.
[6] Alfred Wegener Institute for Polar and Marine Research, Am Alten Hafen 26, D-27568, Bremerhaven, Germany.

*Correspondence to*: Mariem Saavedra-Pellitero (msaavedr@uni-bremen.de)

**Abstract:**

Coccolithophores are globally distributed microscopic marine algae that exert a major influence on the global carbon cycle through calcification and primary productivity. There is recent interest in coccolithophore polar communities, however field observations regarding their biogeographic distribution are scarce for the Southern Ocean. This study documents the latitudinal variability in the coccolithophore assemblage composition and the coccolith mass variation of the ecologically dominant *Emiliania huxleyi* across the Drake Passage. Ninety-six water samples were taken between 10 and 150 m water depth from 18 stations during POLARSTERN Expedition PS97 (February-April, 2016). A minimum of 200 coccospheres per sample were classified in scanning electron microscope and coccolith mass was estimated with light microscopy. We find that coccolithophore abundance and diversity decrease southwards marking different oceanographic fronts as ecological boundaries. We characterize three zones: (1) the Chilean margin, where *E. huxleyi* type A (normal and overcalcified) and type R are present; (2) the Subantarctic Zone (SAZ), where *E. huxleyi* reaches maximum values of $212.5*10^3$cells/L and types B/C, C, O are dominant. (3) The Polar Front Zone (PFZ), where *E huxleyi* types B/C and C dominate. We link the decreasing trend in *E. huxleyi* coccolith mass to the poleward latitudinal succesion from type A to type B group. Remarkably, we find that coccolith mass is strongly anticorrelated to total alkalinity, total $CO_2$, bicarbonate ion and pH. We speculate that low temperatures are a greater limiting factor than carbonate chemistry in the Southern Ocean. However, further in situ oceanographic data is needed to verify the proposed relationships. We hypothesize that assemblage composition and calcification modes of *E. huxleyi* in the Drake Passage will be strongly influenced by the ongoing climate change.



## 1. Introduction

The carbon chemistry of the ocean has a fundamental impact on marine life. The current influx of anthropogenic $CO_2$ into the surface ocean is causing a substantial substantial perturbation to marine chemistry, as exhibited by variations in alkalinity, carbonate ion, saturation state or pH  (e.g., Gattuso et al., 2011). Many organisms use dissolved carbon for photosynthesis and/or the production of calcium carbonate biominerals. Open ocean phytoplankton include coccolithophore algae, an unicellular organism belonging to the phylum Haptophyta (Young and Bown, 1997; Young et al., 2003). Within a single

coccolithophore cell there are dual pathways of carbon utilization – for photosynthesis and for biomineralization of calcium carbonate platelets, called coccoliths. As primary producers of up to ~40% of open ocean calcium carbonate (Poulton et al., 2013) and ~20% of global net marine primary production (Malone et al., 2017), coccolithophore responses to changing ocean chemistry is therefore key for marine ecology and carbon cycling. The ratio between particulate organic carbon formed during photosynthesis and particulate inorganic carbon produced via calcification varies depending on species or even morphotypes

within species (Blanco-Ameijeiras et al., 2016), but can also be highly influenced by environmental conditions, such as seawater $CO_2$ concentration, total alkalinity, and phospahe concentration (Findlay et al., 2011). As climate varies, it is expected that these key conditions will change, and it is expected that upper oceans may experience increased stratification and decreased nutrient availability in the upper photic zone (Cabré et al., 2015). How exactly coccolithophores will respond to these changes is subject to debate.

Ocean acidification combined with the increase in sea-surface temperature due to global warming  are major concerns in polar and subpolar regions (e.g., Wassmann et al., 2011; Post et al., 2013; Freeman and Lovenduski, 2015), triggering an increasing interest in coccolithophore ecology at high latitudes (e.g., Harada et al., 2012; Dylmer et al., 2013; Balch et al., 2016; Charalampopoulou et al., 2016; Giraudeau et al., 2016; Saruwatari et al., 2016; Nissen et al., 2018; Rigual Hernández et al.,

2018; Krumhardt et al., 2019). Questions remain about how coccolithophore populations will adapt to predicted changes in their environment, if at all. There is growing concern that increasing levels of $CO_2$ in the atmosphere and the subsequent acidification of the ocean may disrupt the production of coccoliths. As more of the water column becomes undersaturated in $CaCO_3$ the future (Fabry et al., 2009), carbonate dissolution will be favored over precipitation and coccolithophores may be less successful in exporting carbon to the deep ocean (Fabry et al., 2008). Additionally, any change in the global distribution

and abundance of coccolithophore species relative to non-calcifying groups of phytoplankton (e.g., naked Haptophyceae cells, diatoms, etc) will have important effects on the biogeochemical cycling of carbon and climatic feedbacks. A known positive correlation exists between surface-ocean carbonate ion concentrations $[CO_3^{2-}]$ and the mean coccolith mass of the associated



Noëlaerhabdaceae assemblage, a family of coccolithophores, which includes the extant species *Emiliania huxleyi* (Beaufort et al., 2011). This correlation is driven by the replacement of more- by less-heavily calcified morphotypes or species with declining $[CO_3^{2-}]$. Although the physiological driver for this strong ecological selective pressure is not known (Beaufort et al., 2011), it may determine Noëlaerhabdaceae biogeography, particularly in high latitudes, both in the past and future (Cubillos

et al., 2007).

Geographical shifts in the occurrence or abundance of coccolithophores and assemblage compositions have been actually observed (e.g., Rivero-Calle et al., 2015; Krumhardt et al., 2016). Repeated sampling in the Australian sector of the Southern Ocean (SO) over the past four decades has shown a dramatic range expansion of *Emiliania huxleyi* south of 60ºS (Cubillos et

al., 2007), where any ocean acidification effect appears outweighed by surface-ocean warming. Other authors also recorded a southward expansion of the habitat of *E. huxleyi* in the SO during the last two decades (Winter et al., 2014), although the actual cause of this latitudinal expanse is still under debate (e.g., Patil et al., 2014; Malinverno et al., 2015). Even with a temperature-driven range expansion of coccolithophores in the SO, surface ocean carbonate chemistry is now capable of exerting a first-order control on the composition of coccolithophore assemblages as well as on calcification by coccolithophores from sub-

specific morphotypes (Cubillos et al., 2007; Mohan et al., 2008) to species levels (Beaufort et al., 2011; Freeman and Lovenduski, 2015).

With significant changes in marine species distributions already occurring, it is crucial to understand the ecosystem structure as well as the potential impact of environmental change on the provision of essential ecosystem services (O'Brien et al., 2016).

In this work, we assess the potential relationship between environmental parameters and the community composition, biogeography and calcification mode of modern high latitude coccolithophore communities across the Drake Passage. Accordingly, we calculated extant coccolithophore species numbers at different stations between 10 and 150 m of the water column, evaluated the coccolith mass variations of *E. huxleyi* and compared with in situ conductivity–temperature–depth (CTD) measurements, carbonate chemistry parameters, as well as to previously published Southern Ocean coccolithophore

and calcification datasets.

## 2. Material and methods

### 2.1. Sample preparation for scanning electron microscope analyses and coccolithophore taxonomical considerations

Ninety-six water samples were taken at 18 Stations located in the southern Chilean continental margin and across the western end of the Drake Passage (Fig. 1) from February to April 2016 during Expedition PS97 (Lamy, 2016). Seawater samples were

obtained at different depths using a rosette sampler with 24 × 12 L Niskin bottles (Ocean Test Equipment Inc.) attached to a CTD Seabird SBE911 plus device (Lamy, 2016). The bottles were fired by an SBE32 carousel. For the study of coccolithophore assemblages, 4 to 7 samples per station, between 10 and 150 m water depth, were chosen. Two litres of water





were filtered onto 0.45 µm pore size Polycarbonte Track-Etch Membrane, air-dried and stored over silica gel. A small part of the filter was cut out, fixed on an aluminum Scanning Electron Microscope (SEM) stub and sputtered with gold/palladium. A specific area of the center of the filter was analysed with Zeiss DSM 940A SEM at the University of Bremen, to determine quantitative cell counts for all morphotypes, species and total coccolithophore abundance at magnifications of 1000x, 2000x

and 5000x when required. A minimum of 200 whole coccospheres per sample were counted and classified following Young et al. (2003), the revised classification of Jordan et al. (2004)(2004) and the electronic guide to the biodiversity and taxonomy of coccolithophores Nannotax 3 (ina.tmsoc.org/Nannotax3/index.html) by Young et al. (2019).

Initially, seven different morphotypes of *Emiliania huxleyi* were distinguished in the study area belonging to two main groups,

types A and B (for further details see Nannotax). These are type A (*huxleyi*), type A overcalcified, type B (*pujosiae*), type B/C, type C (*kleijneae*), type R and type O (which included specimens with an opened central area and specimens with the central area covered by a thin plate) (Table 1, Plate 1). Additionally, the degree of calcification was visually assessed while counting, that is why the terms "normal", "calcified" and "heavily calcified" are used in this work to denote some of the most robust *E. huxleyi* placoliths regardless the morphotype (see Plate 1).Semiquantitative estimates of preservation were based on SEM

observations on the coccolithophore assemblage. "Good" preservation implied little or no evidence of carbonate dissolution. Coccoliths with the main morphological characteristics partially altered but still identifiable at species level were tagged as "moderate" (e.g., T-elements within the taxon *E. huxleyi* were present). Specimens affected by strong dissolution or high fragmentation were regarded as "poor" (e.g., T-elements within the taxon *E. huxleyi* were dissolved).

**2.2. Oceanographic data**

The CTD-rosette hydrocasts during Expedition PS97 (Lamy, 2016) provided vertical water column profiles of in situ sea surface temperature (SST), salinity (SSS), density, oxygen and fluorescence (reflecting chlorophyll-α concentrations) (Fig. 2). According to the criteria specified by Orsi et al. (1995), different oceanographic fronts were crossed in this latitudinal transect, the Subantarctic Front (SAF) between stations PS97/036-1 and PS97/037-1 and the Antarctic Polar Front (PF), at PS97/040-1

(Fig. 1). The areas between these fronts are referred to as Subantarctic Zone (SAZ), north op the SAF, Polar Front Zone (PFZ), between the SAF and PF and Antarctic Zone (AZ), south of the PF.

Phosphate, nitrate and silicate contents were retrieved from the World Ocean Atlas 2013 (WOA13) 1°x1° grid austral summer collection (December to February) (Garcia et al., 2014). Total carbon dioxide ($T_{CO2}$) and total alkalinity ($T_{ALK}$) values were

30 obtained from the Ocean Data View (ODV) global alkalinity and total dissolved carbon 1°x1° grid collection from the uppermost 150 m of the water column during austral summer (December to February) (Goyet et al., 2000). Since carbonate chemistry parameters were not measured in situ and data availability in the Drake Passage/Southern Ocean is limited, bicarbonate ion ($HCO_3^-$), carbonate ion ($CO_3^{2-}$), saturation state ($\Omega_{Ca}$) and pH were calculated (as derived– variables) with





ODV software version 4.6.3 (Schlitzer, 2015) using the uppermost 150 m of the GLODAPv2 collection (Key et al., 2015) and excluding measurements done before 1980. $HCO_3^-$, $CO_3^{2-}$, $\Omega_{Ca}$ and pH were also calculated (for comparison purposes) using the CO2SYS.XLS program for 10-20 m depth (Pierrot et al., 2006) and considering the interpolated values of $T_{ALK}$, $T_{CO2}$ and nutrients as well as the in situ measurement of temperature, salinity and pressure.

The values of each of these oceanographic parameters were estimated at the location of every CTD station by interpolating the available data points. The interpolation used a triangulation-based linear method for CTD stations located within the boundaries of the available data, and a nearest neighbour extrapolation method for any CTD stations located outside of the data boundaries. Latitude/longitude coordinates were projected to Universal Transverse Mercator (Zone 19E, World Geodetic

System 1984) before interpolation in order to minimize the distance distortion inherent in geographic coordinates. The calculations were done in MATLAB$^{TM}$ using a custom function that is provided as supplementary material.

### 2.3. Sample preparation and coccolith calcite estimates in light microscope

Fifteen samples were selected in a latitudinal transect for coccolith calcite estimates, between 10 and 20 m water depth. Sample preparation was designed and carried out at the University of Salamanca (Spain). A part of the filter (ca. 1/4 of the original

Polycarbonte Track-Etch Membrane) was cut out and carefully placed into small plastic bags. Buffered water (pH = 9) was prepared with 0.075g/L of $Na_2CO_3$, 0.1 g/L and 0.04g/L of unflavored gelatin ("Gold Gelatin"). Six hundred μL of buffered water were added, and the bags were sealed consistently in a triangular shape. After 30 minutes, the bags were shaken in a lab vortex for 3 minutes to ensure that coccoliths were fully detached from the filter and re-suspended. Holding each bag on top of a cover slide placed on a hot plate (ca. 60°C), an incision was made with a scalpel in one of the bag angles, allowing the

solution to slowly drop onto the cover slide. Once the water was evaporated, the cover slip was mounted with mixture of 50% Canada balsam and 50% Xylene and left in an oven (40° C) for at least 24h. This technique ensured that coccoliths were in the same plane of focus for polarized light microscopy. The dried filter was checked for presence/absence of coccoliths later on in the SEM of the University of Bremen (see supplementary material).

In this study, a Nikon Eclipse LV100 POL polarized light microscope with a 100x H/N2 objective set-up with circular polarization was used at the University of Salamanca. In order to determine coccolith mass and thickness, between 20 and 53 random fields of view were imaged using the Nikon DS-Fi1 digital 8-bit colour camera and the NisElements software, keeping the light level of the microscope and aperture settings constant. The images were processed with the C-Calcita software (for further details see Fuertes et al., 2014). Calibration was done with images of a well preserved and in-focus calcareous spine

from the sample PS97/033-1 at 10 m (see supplementary material). In total, 796 coccoliths were analysed, with a minimum of 34 coccoliths (up to 94) measured per sample.



### 2.4. Diversity indices

The Shannon index (*H*), the Simpson diversity index (1-*D*) and the Fisher's alpha index were calculated with Paleontological Statistics (PAST™) software version 3.22 (Hammer et al., 2001) using the raw coccolithophore counts. *H* was determined with the following equation (1):

$$-\sum_i \frac{n_i}{n} \, ln \frac{n_i}{n}$$
(1)

where $n_i$ is the number of individuals in taxon *i*, and *n* is the total number of all individuals. This index takes into account the number of individuals as well as the number of taxa. *H* ranges from 0 for communities with just a taxon to higher values for communities with many taxa, each with few individuals (Harper, 1999). Dominance, *D*, was determined with the equation (2):

$$D = \sum_i \left(\frac{n_i}{n}\right)^2$$
(2)

The Simpson index 1-*D* varies from 0 to 1 and measures evenness of the community. Fisher's alpha was calculated with the following equation (3):

$S=a*\ln(1+n/a)$
(3)

where *S* is number of taxa, *n* is number of individuals and *a* is the Fisher's alpha.

### 2.5. Statistics

A principal component analysis (PCA) was performed on the coccolithophore relative abundance data using PAST™ software version 3.22 (Hammer et al., 2001). The objective of the PCA is to find hypothetical variables, called components, that capture the maximum proportion of the variance in the multivariate dataset as possible (Davis, 1986; Harper, 1999). These new variables are linear combinations of the original variables (Hammer et al., 2001). The first principal component has the largest

variance possible, and each subsequent component explains the next greatest variance possible. Samples in which less than 50 coccospheres were counted were excluded from the original database, therefore just 74 samples were considered for the PCA. In order to avoid skewness, the relative abundances of the different coccolithophore taxa (x) were log-transformed prior to the PCA using the formula: y=log(x+1). This transformation enhanced the importance of rare taxa, and minimized the dominance of few abundant taxa (Mix et al., 1999), in this case of *E. huxleyi* types B/C and C.

Seventeen coccolithophore taxa were considered for the PCA (see the taxonomical groups in Table 2). Due to the similar ecological preferences observed, *Papposphaera* sp and *Pappomonas* spp. were lumped together in one taxonomical group, in the same way as holococcolithophores, the *Syracosphaera* and the *Ophiaster* species (Table 2). In contrast, *E. huxleyi* morphotypes were regarded as different groups. A correlation matrix between the principal component scores and the

environmental variables (i.e., SST, SSS, fluorescence, oxygen and density measured in situ, as well as nitrate, phosphate and silicate contents interpolated from the WOA13) was performed in order to identify potential relationships between the environmental parameters and the coccolithophore components.



## 3. Results

### 3.1. Coccolithophore distribution

The analysis of 96 water samples shows in general higher cell concentrations in the uppermost 100 m of the water column in the SAZ and at shallower depths (ca. 60 m) in the PFZ (Fig. 3 a). The highest coccolithophore number, $214.6*10^3$cells/L, is reached at station PS97/034-2 (60 m) in SAZ, but high cell concentrations are generally observed north of the SAF (Fig. 3 a). The uppermost 150 m average of coccolithophore concentrations drastically drop at the oceanographic fronts, from $119.9*10^3$ to $56.4*10^3$cells/L (PS97/036-1, /037-1) at the SAF, and from $32.2*10^3$ to $0.1*10^3$ cells/L (PS97/040-1, /041-1) at the PF. Stations south of the PF show very low cell numbers or are devoid of coccolithophores, but detached coccoliths are occasionally observed at the southernmost locations, even at station PS97/50-2. Coccolithophore preservation is in general moderate to good north of the PF, with optimum values at 10-20 m, but becomes notably poorer in the AZ (Fig. 3 c).

### 3.2. Coccolithophore community composition

Twenty-three different coccolithophore taxa (including morphotypes) are observed in this transect across the Drake Passage (Table 2). The most dominant species is *E. huxleyi*, although less abundant taxa also dwell in these (sub-) polar waters. In the following lines, we will comment on the main species composing the coccolithophore community, from the dominant taxa to the rare ones.

#### 3.2.1. *Emiliania huxleyi*

*Emiliania huxleyi* dominates the coccolithophore assemblage, reaching values up to $212.5*10^3$cells/L at PS97/034-2 (60 m). This taxa is present in all the stations, except at coccolithophore-barren samples south of the PF, where coccospheres are occasionally recorded in low numbers, always below $3.0*10^3$cells/L. Seven well established morphotypes of *E. huxleyi* are observed; these are types A, A overcalcified, B, B/C, C, R and O (Plate 1), grouped into A and B (Young et al., 2019). Neither malformed *E. huxleyi*, morphotype D, nor var. *corona* are present in the studied samples. However, a variation in the degree of coccolithophore calcification is observed; i.e., heavily calcified specimens as well as weakly calcified specimens are present in this transect (Table 1, Plate 1).

*Emiliania huxleyi* A group is less abundant than group B and includes type A, type A overcalcified and type R, all of them present in coastal waters (Fig. 4 a). Type R is the most uncommon *E. huxleyi* morphotype, and it just dwells offshore Chile and at station PS97/038-1 (20 m), where it reaches a maximum of $0.9*10^3$cells/L (Fig. 4 c). A similar distribution pattern is shown by *E. huxleyi* type A overcalcified (Fig. 4 d), with concentrations up to $1.8*10^3$cells/L at PS97/018-1 (100 m). *Emiliania*



*huxleyi* type A (normal form or moderately calcified) is restricted to the continental margin, where it records numbers of up to 1.8*10³cells/L (Fig. 4 b).

*Emiliania huxleyi* B group specimens dwell north of the PF, showing a broader distribution than A group (Fig. 5 a); it includes

types B, B/C, C and O. Type B is the most unusual *E. huxleyi* morphotype within group B, with maximum numbers of 7.4*10³cells/L at PS97/029-1 (10 m). *Emiliania huxleyi* type B is abundant in a narrow deep band (0-150 m) at the Chilean margin/open ocean transition in the SAZ, and it is occasionally present in shallow waters of the PFZ (Fig. 5 b). Types B/C and C show a broader latitudinal distribution and they are the most abundant *E. huxleyi* taxa with maxima of 74.0*10³cells/L at PS97/030-1 (150 m) and 103.2*10³cells/L at PS97/034-2 (60 m), respectively (Table 2, Fig. 5 c, d). Type B/C reaches higher

concentrations at shallower depths (10-20 m), when compared to type C (20-100 m). Type O, with an "opened" central area, displays a similar distribution pattern to type B/C and reaches the highest numbers (8.0*10³cells/L at PS97/032-1, 10 m) mainly at shallow depths (10-20 m) (Fig. 5 e). On the contrary, type O with a central area covered by a thin "lamella", which is highly variable in size, is notably more abundant and dwells at deeper depths (ca. 0-100 m), with maximum abundances of 72.0*10³cells/L at PS97/036-1, 60 m (Fig. 5 f). *Emiliania huxleyi* detached coccoliths were not counted but their presence or

absence was assessed during the SEM analyses. Free detached coccoliths of *E. huxleyi* types B/C-C show a broader distribution than coccospheres and are rarely recorded up to 61.7ºS at station PS97/050-2.

### 3.2.2. Other taxa

On top of *E. huxleyi*, less abundant taxa are observed. *Ophiaster* spp., including *O. hydroideus* (Plate 2) and sporadically *O. reductus*, is present primarily at the Chilean margin to a maximum 150 m water depth, but unexpectedly reaches up

10.1*10³cells/L at PS97/038-1 at a depth of  10 m north of the PF (Fig. 6 a). *Calciopappus caudatus* is found in the uppermost 60 m of the SAZ and PFZ with maximum abundances of 5.3*10³cells/L also at PS97/038-1, 10 m (Fig. 6 b). Four species belonging to the genus *Syracosphaera* are recorded in the Drake Passage (i.e., *S. dilatata*, *S. corolla*, *S. marginaporata* and *S. pulchra*). *Syracosphaera* spp. dominates in the SAZ, except at coastal stations, and its highest numbers (up to 5.3*10³cells/L PS97/038-1) are recorded between 10 and 60 m (Fig. 6 c). In contrast, *Calcidiscus* s.l., mainly *Calcidiscus leptoporus*, displays

moderate numbers at the coastal stations offshore Chile (from 10 to 150 m), but reaches higher numbers (up to 1.4*10³cells/L at PS97/038-1, 10 m) southwards in a rather patchy, but shallow distribution (Fig. 6 d). *Papposphaera* and *Pappomonas* spp. (including sp. 1 and sp. 5) are observed mainly in the PFZ or at shallow depths in the SAZ, with a maximum concentration of 1.4*10³cells/L at PS97/038-1, 10 m (Fig. 6 e). Holococcolithophores, mainly *Syracosphaera strigilis* HOL (Plate 2), are restricted to the uppermost 60 m in the SAZ with maximum values of 1.0*10³cells/L (PS97/038-1) (Fig. 6 f).

Rare coccolithophore taxa (maximum below 0.8 *10³cells/L) are also recorded in the Drake Passage. *Gephyrocapsa muellerae* is restricted to the northernmost stations offshore of Chile, while *Chrysotila* sp. is occasionally observed in the SAZ and *Calciosolenia murrayi* in the PFZ (Fig. 7 c-e). *Acanthoica quattrospina* and *Wigwamma antarctica* display a broader





distribution, even south of the PF, in the case of the later (Fig. 7 a, b). Additionally, non-coccolithophores haptophytes belonging to the genera *Petasaria* and *Chrysochromulina* are generally present in the SAZ, PFZ and AZ, reaching maximum concentrations of 7.8 $*10^3$cells/L at PS97/037-1 at 60 m water depth (Fig. 7 f).

In order to investigate the relationship between the coccolithophore composition and the environmental variables in the study area, a PCA, was performed using the data from 74 sampling points (Fig. 8). Based on the broken stick method, the PCA indicates the existence of three main principal components (PC) explaining the 77.4% of the total variance (see supplementary material). PC1 explains 45.3% of the variance and it is positively related to the abundance of *E. huxleyi* type O and negatively related to *E. huxleyi* type C. PC2, connected to *E. huxleyi* type B/C, in a lesser extent to type B, and negatively correlated to

*E. huxleyi* type A, accounts for 17.3% of the variance. PC1 seems associated to a marked temperature gradient. Positive PC1-values are linked to warm/moderately warm water taxa, which dwell north of the SAF, and negative PC1-values are connected to taxa that live in colder waters at higher latitudes. PC1 is correlated to SST and anticorrelated to density, oxygen content, macronutrients (phosphate, nitrate) and silicate (Table 3). PC2 is related to salinity variations (Table 3). Negative values of PC2 are related to extant coccolithophore taxa observed in the low salinity waters offshore Chile or to taxa living in the PFZ.

On the contrary, positive values of PC2 are linked to taxa dwelling in the SAZ, where the SSS values are the highest of the studied transect. The coccolithophore assemblage composition separates out three different clusters in the PCA (Fig. 8) corresponding to three different oceanographic areas. The SAZ is characterized by positive values PC1 and PC2, the PFZ by negative PC1 and PC2 values, and the Chilean coastal envirorment (Chl) by positive PC1 and negative PC2.

### 3.3. Diversity

Coccolithophore diversity indices (Shannon index *H*, Simpson index 1-D, Fisher's alpha) as well as the number of taxa are highest offshore Chile (0-150 m) and they became restricted to the uppermost 60 m in the PFZ (Fig. 9). The number of taxa is maximum at station PS97/038-1 (10 m) and drastically drops south of the PF. The Shannon and Simpson 1-D indices display a similar pattern (Fig. 9 b, d), are highly correlated (r=0.95, supplementary material) and show that coccolithophore diversity decreases southwards. The number of taxa and diversity indices are strongly related to latitude (Fig. 9 c, f), with station

PS97/038-1 (10-20 m) being an outlier due to the high diversity estimates recorded there.

### 3.4. Coccolith calcification –*E. huxleyi*-

*Emiliania huxleyi* specimens were classified into non-standardized sub-categories (e.g., regarding level of coccolith calcification) while counting in the SEM. Specimens of type A calcified/heavily calcified and overcalcified (Fig. 10 a) are present at the same locations as the common *E. huxleyi* type A (Fig. 4). Calcification in the group B implied in most of the

cases a thicker central tube in the coccolith central area and thicker T-elements (Plate 1). Although type B calcified is recorded with rather low numbers, it shows a similar distribution to type B/C calcified (Fig. 10 b, c); both are restricted to the uppermost



150 m of the water column at stations PS97/029-1, /030-1 and /031-1. Type C calcified is occasionally recorded where types B and B/C are present, but shows a much broader and patchy distribution north of the PF (Fig. 10 d).

Coccolith mass measured with the software C-Calcita ranges from 19.8 to 0.8 picograms, and median values (per station) vary from 7.3 to 2.4 pg. Relatively high *E. huxleyi* masses are recorded in the SAZ (Fig. 11), but not at the stations with the highest cell concentrations (such as PS97/034-2). Measurements of the coccolith mass allowed us to compare to the identified morphotypes. In general, coccolith mass decreases southwards across the Drake Passage (Fig. 11). While high coccolith masses are reached offshore Chile (i.e., stations PS97/017-1, /018-1) where *E. huxleyi* types R and A (overcalcified) are present, low coccolith masses are reached in the PFZ where types B/C and C dominate (i.e., stations PS97/037-1, /040-1). The gradual latitudinal mass decrease is occasionally interrupted by sudden drops in the mass estimates. These decreases in mass estimates appeared to be controlled by the predominance of a specific morphotype, for instance the low *E. huxleyi* mass values recorded at PS97/016-1 coincide with a sudden increase in the relative abundance of *E. huxleyi* type C (Fig. 11).

## 4. Discussion

### 4.1. Latitudinal variations in the coccolithophore abundance, distribution and diversity.

The observed maximum coccolithophore abundance recorded (up to 214.6*10³cells/L) is in agreement with previous studies carried out in different sectors of the Southern Ocean, which estimated maximum numbers between 130 and 640*10³cells/L (e.g., Eynaud et al., 1999; Findlay and Giraudeau, 2000; Cubillos et al., 2007; Gravalosa et al., 2008; Mohan et al., 2008; Hinz et al., 2012; Saavedra-Pellitero et al., 2014; Malinverno et al., 2015; Balch et al., 2016; Charalampopoulou et al., 2016). The coccolithophore abundance and diversity drastically drop from North to South and portrays the oceanographic fronts as ecological boundaries. Marked shifts in the coccolithophore numbers, community composition and diversity occurring at the SAF and PF observed here, were also previously noted by other authors in different sectors of the Southern Ocean (e.g., Eynaud et al., 1999; Gravalosa et al., 2008; Saavedra-Pellitero et al., 2014; Malinverno et al., 2015; Balch et al., 2016; Charalampopoulou et al., 2016). Although the aforementioned studies reported increases in the abundance of coccolithophores at the SAF and PF, we only observe an increase in the number of cells/L in the PF at shallow depths (< 60 m, Fig. 3), which is not so evident in the SAF. The increase in coccolithophore abundance recorded in the PF could be linked to the high biological productivity ocurring at the antarctic circumpolar current fronts (e.g., Murphy, 1995; Pollard et al., 2002; Patil et al., 2013) due to the frontal dynamics itself (e.g., Laubscher et al., 1993) or to the physical accumulation of particulate matter and nutrients at these convergence zones (e.g., Franks, 1992; Eynaud et al., 1999; Gravalosa et al., 2008; Balch et al., 2016).

The southernmost extent of *E. huxleyi* has been extensively discussed (e.g., Winter et al., 2014; Malinverno et al., 2015). The PF constitutes a natural sharp barrier which marks a drop in coccolithophore diversity and number of coccospheres (Saavedra-Pellitero et al., 2014; Saavedra-Pellitero and Baumann, 2015). Several studies observed the absence of *E. huxleyi* south of the





PF (e.g., Verbeek, 1989; Charalampopoulou et al., 2016). However, specimens of *E. huxleyi* and *W. antarctica* are sporadically recorded south of the PF in the studied transect with numbers of <3*10³cell/L in the uppermost 80 m of the water column at stations PS97/043-3 and /047-1. *Emiliania huxleyi* is observed in low numbers at temperatures between 1.7 and -0.7ºC (Fig. 2), below the 2º C isotherm limit that McIntyre and Bé (1967) originally established for the Atlantic Southern Ocean. Although

it is unusual, few authors occasionally found *E. huxleyi* also dwelling in cold waters < 2º C (see Table 1 in Holligan et al., 2010). Monospecific assemblages of *E. huxleyi* have been also recorded south of the PF by other authors in the Pacific sector (e.g., Gravalosa et al., 2008; Saavedra-Pellitero et al., 2014), Australian sector (e.g., Nishida, 1986; Findlay and Giraudeau, 2000; Cubillos et al., 2007; Malinverno et al., 2015), in the Atlantic sector (e.g., Eynaud et al., 1999; Holligan et al., 2010) and Indian sector (e.g., Mohan et al., 2008; Patil et al., 2014). We speculate that the free detached coccoliths of *E. huxleyi* observed

in our study area, up to 61.7 ºS, and showing a broader distribution than coccospheres, are not in situ and could have been transported. In any case the southernmost extent of coccolithophores is also influenced by the clear dominance of diatoms south of the PF, as suggested by the high diatom concentration (valves/g dry sediment) and biogenic opal content recorded in surface sediment samples from the same study area in the AZ (Cárdenas et al., 2018).

The number of taxa and coccolithophore diversity decreases southwards (Fig. 9) in agreement with other studies performed in the Drake Passage, the Australian and Pacific sectors of the Southern Ocean (e.g., Findlay and Giraudeau, 2000; Gravalosa et al., 2008; Saavedra-Pellitero et al., 2014; Charalampopoulou et al., 2016). Coccolithophore diversity is related to the temperature gradient, as shown by the correlation between Shannon index and SST (r=0.8, see supplementary material). Contrary to Saavedra-Pellitero et al. (2014), the highest coccolithophore diversity values do not always occur at stations that

showed the highest coccolithophore abundances (Figs. 3, 9). Few studies offshore Chile (ca. 33ºS, 36 ºS) and in the Drake Passage showed low costal coccolithophore diversity increasing towards oceanic regions (Charalampopoulou et al., 2016; Menschel et al., 2016; von Dassow et al., 2018) which contrasts with the high number of taxa recorded in this work at uppermost 100 m of the water column at the Chilean margin (Fig. 9). Amongst different environmental factors, temperature could be one of the main variables favouring high coccolithophore diversity at the coastal stations.

The unexpected high diversity and number of taxa recorded at 10-20 m at station PS97/038-1 (previously labelled as "outlier"), coincident with relatively high density of coccospheres, does not seem to have been promoted by high SSTs, but rather by an occasional variation in the nutrient availability. So far, there are no nutrient measurements in situ available for this transect, but the interpolated values from the WOA13 austral summer suggest that nutrients are generally available this part of the SAZ,

and shallow (10, 20 m) nitrate or phosphate concentrations (Fig. 12 h, i) do not abruptly change at PS97/038-1. Therefore this could be due to meseoscale eddies which could have advected nutrients (Frenger et al., 2018). We speculate that an increase in the fluorescence values (Fig. 2), reflecting higher chlorophyll-α concentrations, could be attributed to a higher availability of nutrients, which could have favored coccolithophores. The available fluorescence data (Fig. 2) seem to primarily reflect diatom concentration, south of the PF, followed by the non-coccolithophore haptophytes (*Petasaria* and *Chrysochromulina*



spp.) in the SAZ and PFZ (up to 100 m water depth) superimposed to the coccolithophore distribution. Future quantitative analyses of extant diatoms and nutrients performed at the same stations and depths, are envisaged and will be required for better understanding of the phytoplankton communities interactions and ecological patterns across the Drake Passage.

**4. 2. Coccolithophore assemblages/Community composition in the study area/across the Drake Passage**

Based on the PCA (Fig. 8) it was possible to distinguish three main different oceanographic areas, characterized by different coccolithophore assemblages in the study area.

(1) The Chilean margin. *Emiliania huxleyi* type A is present in the stations closest to the Chilean coast (i.e., PS97/018-1 and /017-1), which recorded the highest SST and lowest SSS in the study area. This morphotype of *E. huxleyi* has been also observed in low abundances north of the SAF in different Pacific/Australian Southern Ocean (e.g., Cubillos et al., 2007; Saavedra-Pellitero et al., 2014; Malinverno et al., 2015), although was not observed by others authors (e.g., Gravalosa et al., 2008; Charalampopoulou et al., 2016), probably due to the high latitudes of those transects. Specimens of type A with different

degree of calcification were present ranging from normal to overcalcified (Plate 1), the latter being more abundant. In the Northern Hemisphere, *E. huxleyi* type A dominates the coccolithophore assemblage in the North Atlantic and in Norwegian coastal waters (e.g., van Bleijswijk et al., 1991; Holligan et al., 1993), but not in the Southern Ocean (e.g., Cook et al., 2011; Hagino et al., 2011). *Emiliania huxleyi* type R was observed for first time in the Drake Passage, although it has been previously just observed in the Eastern South Pacific (Beaufort et al., 2008; Beaufort et al., 2011; von Dassow et al., 2018).

Other minor taxa present in the Chilean margin are *Calcidiscus* s.l., *Ophiaster* spp and *G. muellerae*. The first has been found by other authors in the SAZ (e.g., Saavedra-Pellitero et al., 2014; Malinverno et al., 2015), but its patchy distribution north of the PF is in agreement with observations by Charalampopoulou et al. (2016). *Ophiaster* spp. shows uneven distribution in the SAZ and PFZ, but it is also present in the Chilean continental margin. *Ophiaster* spp. is recorded offshore Chile with low

number living up to 100 m water depth, which contrast the high numbers observed in the SAZ offshore New Zealand always above 60 m (Saavedra-Pellitero et al., 2014). Occasional low numbers of extant *G. muellerae* have also been observed in the Southern Ocean by Saavedra-Pellitero et al. (2014) and Findlay and Giraudeau (2000).

(2) The SAZ. This oceanographic zone is bounded in the south by the SAF. Salinity values are relatively constant at about 34

30 psu, but SST gradually decreases, up to ca. 6º C, while the nitrate and phosphate contents progressively increase (Figs. 2, 12). The SAZ is characterized by the dominance of *E. huxleyi* types C, B/C, O and B. The shift in occurrence from type A group to type B group has been recorded by some authors at the STF in the Australian sector (e.g., Hiramatsu and De Deckker, 1996; Findlay and Giraudeau, 2000; Malinverno et al., 2015). *Emiliania huxleyi* type B appears in very low abundance, in agreement



with Saavedra-Pellitero et al. (2014), restricted to a transitional narrow deep band (0-150 m) between the Chilean coastal margin and more open conditions of the SAZ. It has been observed also north of the STF in the Indian Southern Ocean (Patil et al., 2013). This morphotype was otherwise only found in the Northern Hemisphere (Cook et al., 2011).

*Emiliania huxleyi* type O was established Hagino et al. (2011), who observed this morphotype extensively distributed in the Southern Ocean. However, so far only Malinverno et al. (2015) and this study described *E. huxleyi* type O. This would mean that the reported geographic distributions of types B/C and C may be biased by inclusion of Type O (Hagino et al., 2011). *Emiliania huxleyi* type O (including the opened and lamella forms) is abundant in the SAZ, which is in agreement with Malinverno et al. (2015). *Emiliania huxleyi* type B/C and C (see Table 1) are the dominant taxa in the SAZ of the Southern

Ocean (e.g., Findlay and Giraudeau, 2000; Cubillos et al., 2007; Gravalosa et al., 2008; Mohan et al., 2008; Saavedra-Pellitero et al., 2014; Saavedra-Pellitero and Baumann, 2015).

Among minor taxa, *Syracosphaera* spp, *Calcidiscus* sp., *A. quattrospina* as well as holococcolithophores are found in the SAZ, in agreement with the assemblage observed by Charalampopoulou et al., (2016) north of the PF. Four species belonging to the

genus *Syracosphaera* (plus *S. strigilis* HOL) are recorded in the study area, as previously observed (Gravalosa et al., 2008; Charalampopoulou et al., 2016). *Acanthoica quattrospina*, a species tolerant to low salinity (Supraha et al., 2014) is present in southern high latitudes (Eynaud et al., 1999; Findlay and Giraudeau, 2000; Malinverno et al., 2015), but has not been recorded in other polar transects (e.g., Mohan et al., 2008).

(3) The PFZ. This oceanographic zone is bounded by the SAF and the PF. Salinity and SST gradually decreases with respect to the SAZ, and nutrient contents continue to progressively increase poleward (Figs. 2, 12). *Emiliania huxleyi* types B/C and C dominate the PFZ and reach relatively high numbers north of the PF, although *E. huxleyi* type O is still present in low abundance (Fig. 11), as also observed by Malinverno et al. (2015). The high coccolithophore numbers observed in the SAZ and at the shallowest depths of the PFZ are part of the Great Calcite Belt, a region of high surface reflectance in the Southern

Ocean due to the increased seasonal concentrations of coccolithophore and particulate inorganic carbon (Balch et al., 2011; Balch et al., 2016). We suggest that the uneven distribution of some of the coccolithophore taxa is driven by the physical processes of the ACC; i.e., it is primarily linked to the positions of frontal boundaries but also affected by the dynamics of mesoscale eddies, as mentioned by Holligan et al. (2010).

Minor taxa present in the PFZ, and broadly north of the PF include species of the family Papposphaeraceae (i.e., *Papposphaera* sp, *Pappomonas* spp. and *W. Antarctica*). They have small-sized and lightly-calcified coccoliths, which makes them easily overlooked even under SEM. Specimens from the genera *Papposphaera* and *Pappomonas* have been observed at polar waters in the North Hemisphere (Thomsen, 1981; Samtleben and Schröder, 1992; Charalampopoulou et al., 2011; Thomsen and Østergaard, 2014) and also in the Southern Ocean (e.g., Gravalosa et al., 2008; Saavedra-Pellitero et al., 2014;





Charalampopoulou et al., 2016). Apart from the fact that the lightly calcified polar coccolithophores are non-photosynthetic heterotrophs, which gives them a strong competitive advantage to dwell in the darkness for months every year, very little is known about them (Thomsen and Østergaard, 2013). However, because they are weakly calcified, they will be one of the first polar taxa to be threatened by ocean acidification (Thomsen and Østergaard, 2013).

### 4.3. *Emiliania huxleyi* mass variations across the Drake Passage.

In this study, the coccolith mass of *E. huxleyi* was measured across the Drake Passage up to the PF at depths ranging between 10 and 20 m water depth. The general southwards decreasing trend in *E. huxleyi* mass (Fig. 11) is in agreement with trends
oberved by Charalampopoulou et al. (2016) across the Southern Ocean (Fig. 13). Differences in the estimated mass values can be attributed to the distinct taxonomical considerations, to the methodologies used in both studies, and mainly to the different oceanographic conditions during the sampling periods (2009, 2016). The mean coccolith mass is related to strong latitudinal gradients in temperature (r=0.75), also observed by Charalampopoulou et al. (2016), total alkalinity (r=-0.89), total $CO_2$ (r=-0.86), $HCO_3^-$ (r=-0.81 or -0.68 depending on the method used to calculate it) in agreement with Beaufort et al., (2011) and
nutrient content (nitrate: r=-0.75, phosphate=-0.71) noted by Charalampopoulou et al., (2016) (Table 4, Fig. 12). In contrast, the coccolith mass relationship to salinity, fluorescence, silicate content, carbonate ion, calcite saturation is not significant (Table 4). Although the pH variation is rather reduced, the anticorrrelation between coccolith mass and pH becomes significant (r=-0.7) depending on the method used (Table 4, Fig. 12), in agreement with the biogeochemistry and optics South Pacific experiment (BIOSOPE) data from Beaufort et al (2011) (r=-0.52). The negative correlation of the present data contrasts with
the global and well established relationship between coccolith mas and pH (r=0.75) (Beaufort et al., 2011). However, the relationship between coccolith mass and the carbonate chemistry parameters should be considered carefully. $T_{ALK}$, $T_{CO2}$, $\Omega_{Ca}$, pH and $HCO_3^-$ have been calculated from the GLODAP-v2 database (in which the majority of the datapoints are scattered and samples were measured just in August 2005 and in February 2009) (Key et al., 2015) or from the global alkalinity and total dissolved carbon collection (Goyet et al., 2000) which shows an austral summer average using the CO2SYS.XLS program
(Pierrot et al., 2006).

The observed decreasing trend of the coccolith mass can be linked to the latitudinal succession from type A group to type B group (Fig 12 b), in agreement with other authors who observed a latitudinal trend from *E. huxleyi* more calcified to weakly calcified morphotypes (e.g., Cubillos et al., 2007; Mohan et al., 2008). In the Chilean margin, the highest coccolith masses
recorded are related to the presence of *E. huxleyi* type A (including normal, calcified and overcalcified specimens) and type R (Fig. 11), observed also by other authors at lower latitudes offshore Chile (Beaufort et al., 2008; Beaufort et al., 2011; von Dassow et al., 2018). Although it is uncommon, heavily calcified *E. huxleyi* morphotypes have been recorded in reduced pH and $\Omega_{Ca}$ conditions in other parts of the globe (e.g., Smith et al., 2012; Triantaphyllou et al., 2018). The presence of calcified





specimens of *E. huleyi* type B/C, and in a lesser extent of type C, in the transitional zone from the chilean margin to the open ocean reflects an increase in the coccolith masses (Figs. 10, 11). In contrast, a higher relative abundance of *E. huxleyi* type C correponds to smaller coccolith masses in the SAZ. Stricking are the relatively low coccolith mass values in the open ocean of the SAZ that coincide with maxima in the density of coccolithophores (Fig. 12).

The dataset presented here constitutes an important contribution to the coccolithophore ecology sparsely studied at high latitudes. This work is also relevant for future climate and ocean model simulations in the context of global warming and ocean acidification threatening calcifying plankton. Taking into account the existing relationships between the physico-chemical parameters and the coccolithophore components, changes in the composition and calcification modes of *E. huxleyi*

morphotypes are expected to occur in the Drake Passage with the ongoing climate change.

### 5. Conclusions/Summary

This study documents the latitudinal variability in the coccolithophore assemblage composition and calcification of *Emiliania huxleyi*, the dominant species, across the Drake Passage, driven by physical, chemical and biological parameters in the surface

ocean. Coccolithophore abundance and diversity decrease southwards portraying the oceanographic fronts as ecological boundaries. Marked shifts in the coccolithophore numbers, community composition and diversity occur at the subantarctic front (SAF) and polar front (PF). Three main different oceanographic areas are characterized, based on the coccolithophore composition:

(1) The Chilean margin. *Emiliania huxleyi* type A (normal and overcalcified) and type R are present in the stations closest to

20 the Chilean coast, which record the highest SST and lowest SSS in the study area. Rare taxa present offshore Chile are *Calcidiscus* s.l., *Ophiaster* spp and *Gephyrocapsa muellerae.*

(2) The Subantarctic zone (SAZ). This zone is bounded by the SAF in the south. Salinity values are relatively constant and SST gradually decreases while nutrient content increases. The SAZ is characterized by the dominance of *E. huxleyi* types C, B/C, O and B. *Emiliania huxleyi* reaches maximum values of $212.5*10^3$cells/L north in the SAZ. Minor taxa include

*Syracosphaera* spp. (including holococcolithophores from this genus), *Calcidiscus* sp. and *Acanthoica quattrospina.*

(3) The polar front zone (PFZ). It is bounded by the SAF in the north and the PF in the south. Salinity and SST progressively decreases with respect to the SAZ, and nutrient contents continue to increase poleward. *Emiliania huxleyi* types B/C and C dominate the PFZ and reach relatively high numbers north of the PF, although *E. huxleyi* type O is still present.

Minor taxa present in the PFZ, and broadly north of the PF include species of the family Papposphaeraceae (i.e., *Papposphaera*

sp, *Pappomonas* spp. and *Wigwamma antarctica*). Specimens of *E. huxleyi* and *W. antarctica* are sporadically recorded south of the PF with numbers of $<3*10^3$cell/L and dwelling a at temperatures < 2° C.



The general decreasing trend in *E. huxleyi* coccolith mass can be linked to the latitudinal succession from type A group (in the Chilean margin) to type B group (in the PFZ). Coccolith mass and coccolithophore diversity are related to the strong latitudinal gradient in temperature. Coccolith mass also shows anticorrelation to total alkalinity, total $CO_2$, bicarbonate ion ($HCO_3^-$), pH and nutrient content, which contrasts with the global and well established positive relationship between coccolith mass and pH

5    as well as total alkalinity. However, the relationship between coccolith mass and the carbonate chemistry parameters should be considered carefully, since in situ measurements are not available. The existing relationships between the physico-chemical parameters and the coccolithophore components in the Drake Passage suggest that assemblage composition and calcification modes of *E. huxleyi* will be strongly affected by the ongoing climate change.

**6. Acknowledgements**

The Alfred-Wegener-Institute Bremerhaven provided the plankton samples required for this study. R/V POLARSTERN officers and crew are thanked for their help during PS97 Expedition. Marius Becker (University of Kiel) is acknowledged for

15    his help with MATLAB™, Jeremy Young (University College London) for his assistance with coccolithophore identification and Chloe Anderson (Marum) as well as Diederik Liebrand (University of Bremen) for their comments on the manuscript. Funding was provided by the Deutsche Forschungsgemeinschaft (DFG), Reference: BA 1648/30-1, to M. Saavedra-Pellitero. Data is available in the Pangaea data repository.



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




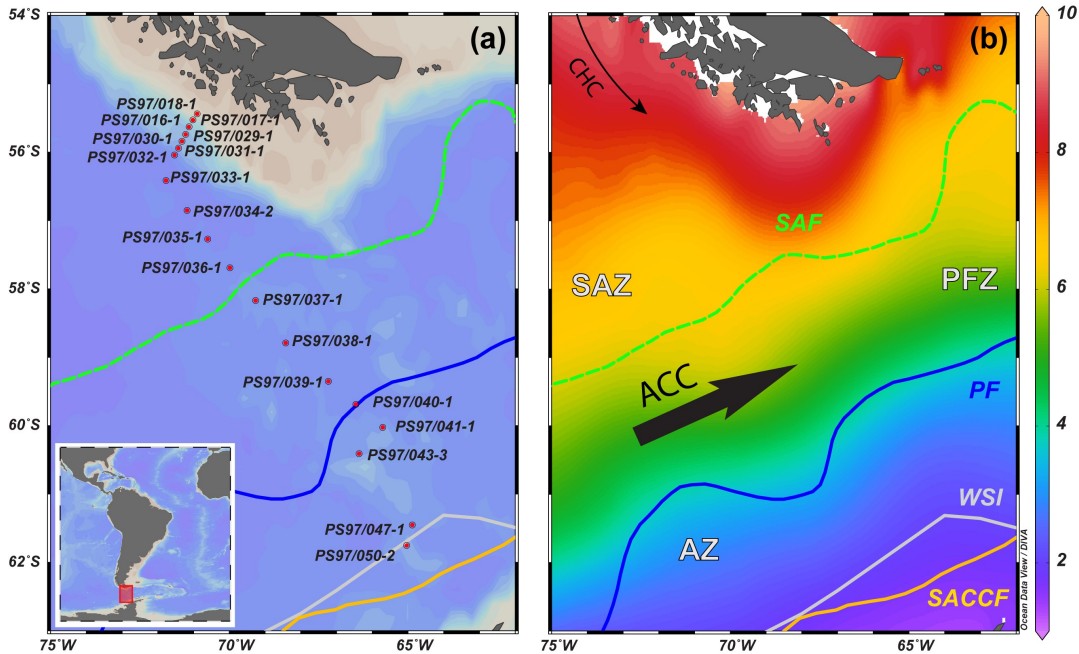

**Fig. 1. (a)** Overview map showing the bathymetry of the Drake Passage area and the CTD stations studied; **(b)** sea surface temperature (°C) seasonal average (January, February, March) at 0 m water depth from the World Ocean Atlas 2013, 0.25° grid (Locarnini et al., 2013) plotted with Ocean Data View (ODV) software version 4.6.3 (Schlitzer, 2015). The different oceanographic fronts are indicated as follows: Subantarctic Front (SAF) with a green dashed line, Antarctic Polar Front (PF) with a blue line and southern Antarctic Circumpolar Current Front (SACCF) with an orange line (Orsi et al., 1995). The areas between the fronts are referred to as Subantarctic Zone (SAZ), Polar Front Zone (PFZ) and Antarctic Zone (AZ). CHC: Cape Horn Current, ACC: Antarctic Circumpolar Current, WSI: winter sea ice extent (Comiso, 2003).




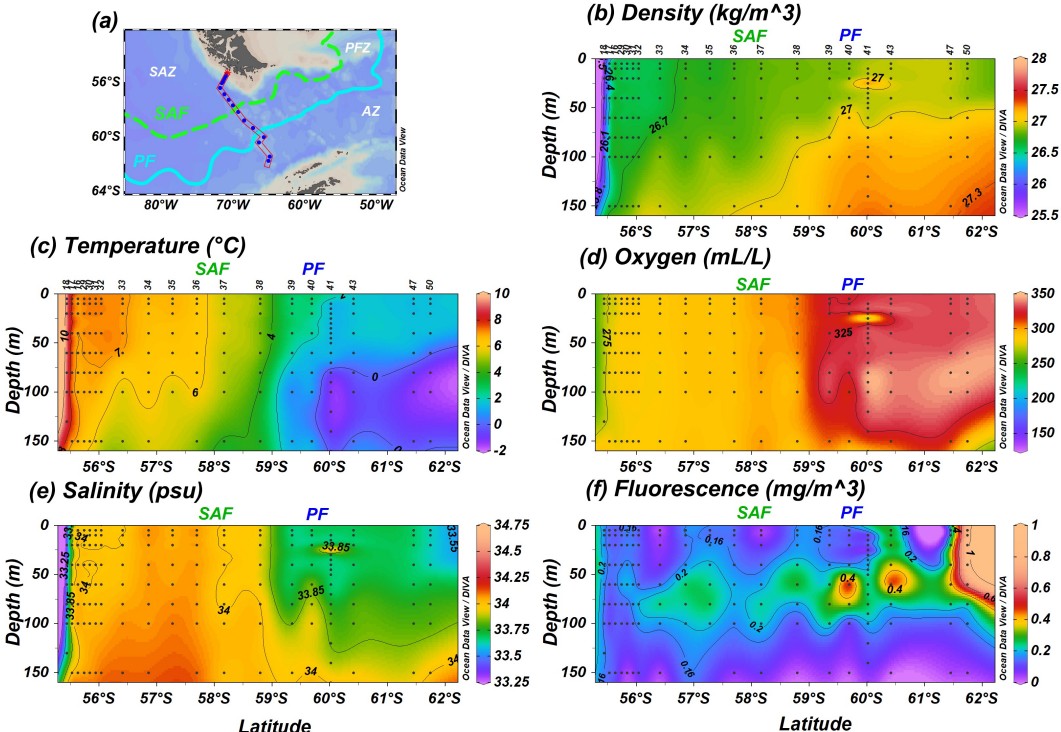

Fig. 2. (a) Location of the PS97 CTD stations studied, (b) water density (kg/m³), (c) sea surface temperature (°C), (d) dissolved oxygen (mL/L) (e) sea surface salinity (psu) and (d) fluorescence (mg/m³) profiles from 55.4°S to 61.7°S up to a depth of 150 m. Black dots indicate sampling points. The different oceanographic fronts (according to Orsi et al., 1995) are indicated as SAF—green dashed line, PF—blue line.



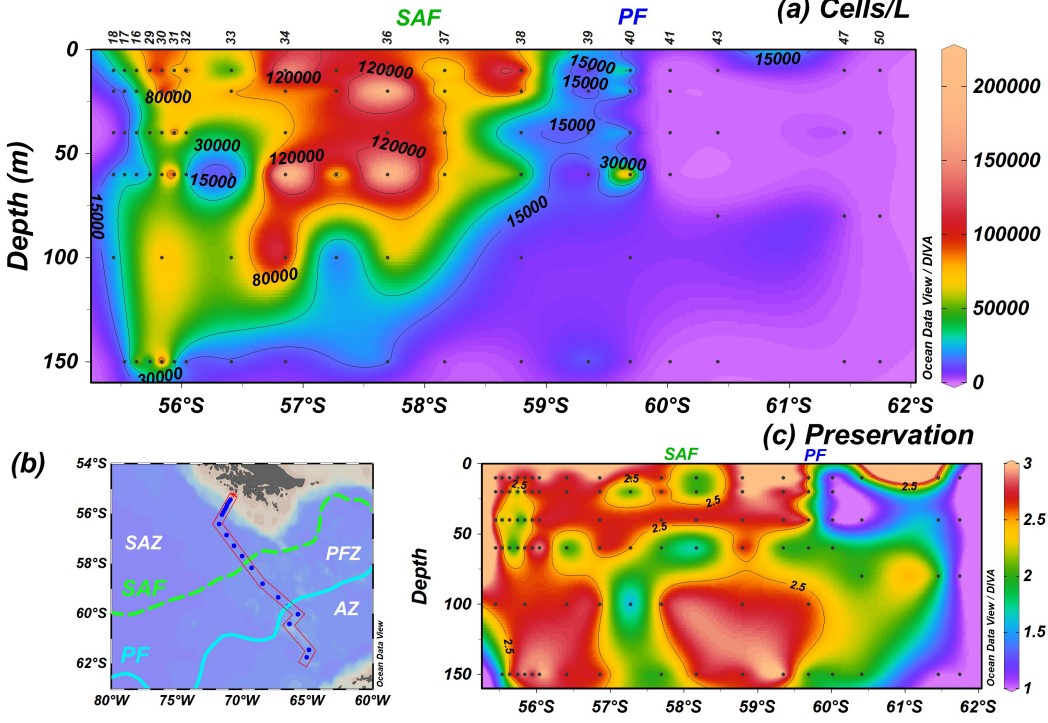

**Fig. 3. PS97 CTD stations showing (a) the number of coccospheres/l across a (b) latitudinal transect in the Drake Passage and (c) semiquatitative estimates of coccolith preservation for the uppermost 150 m of the water column (1=poor, 2=moderate, 3=good). Black dots indicate sampling points and the different oceanographic fronts according to Orsi et al., (1995) are indicated as SAF—green dashed line, PF—blue line.**





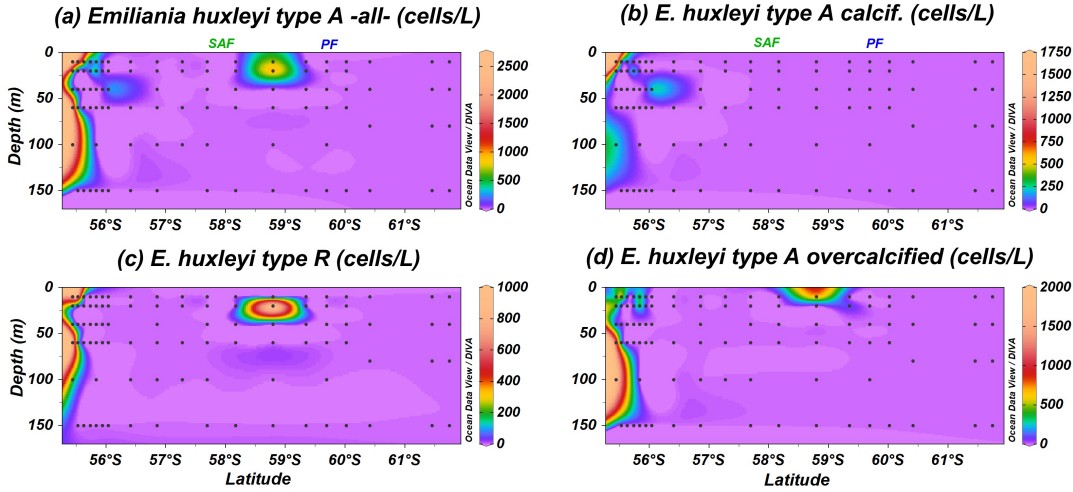

Fig. 4. Location of the PS97 CTD stations studied showing the number of coccospheres/L of (a) *Emiliania huxleyi* type A group, (b) *E. huxleyi* type A moderately calcified, (c) *E. huxleyi* type R and (d) *E. huxleyi* overcalcified. The Subantarctic Front (SAF) and the Polar Front (PF) according to Orsi et al. (1995) are indicated.

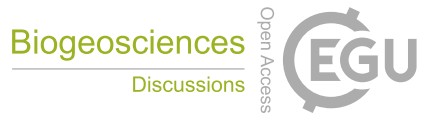

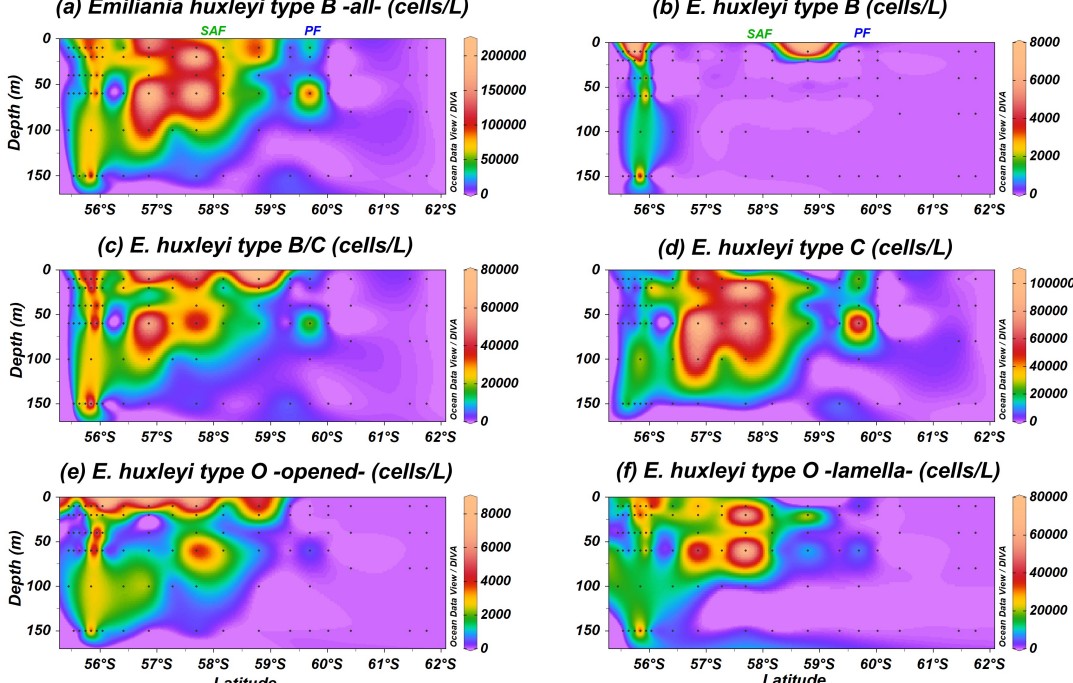

**Fig. 5. Location of the PS97 CTD stations studied showing the number of coccospheres/L of (a)** *Emiliania huxleyi* **type B group, (b)** *E. huxleyi* **type B, (c)** *E. huxleyi* **type B/C, (d)** *E. huxleyi* **type C, (e)** *E. huxleyi* **type O (with opened central area) and (f)** *E. huxleyi* **type O (with lamella). The Subantarctic Front (SAF) and the Polar Front (PF) according to Orsi et al. (1995) are indicated.**





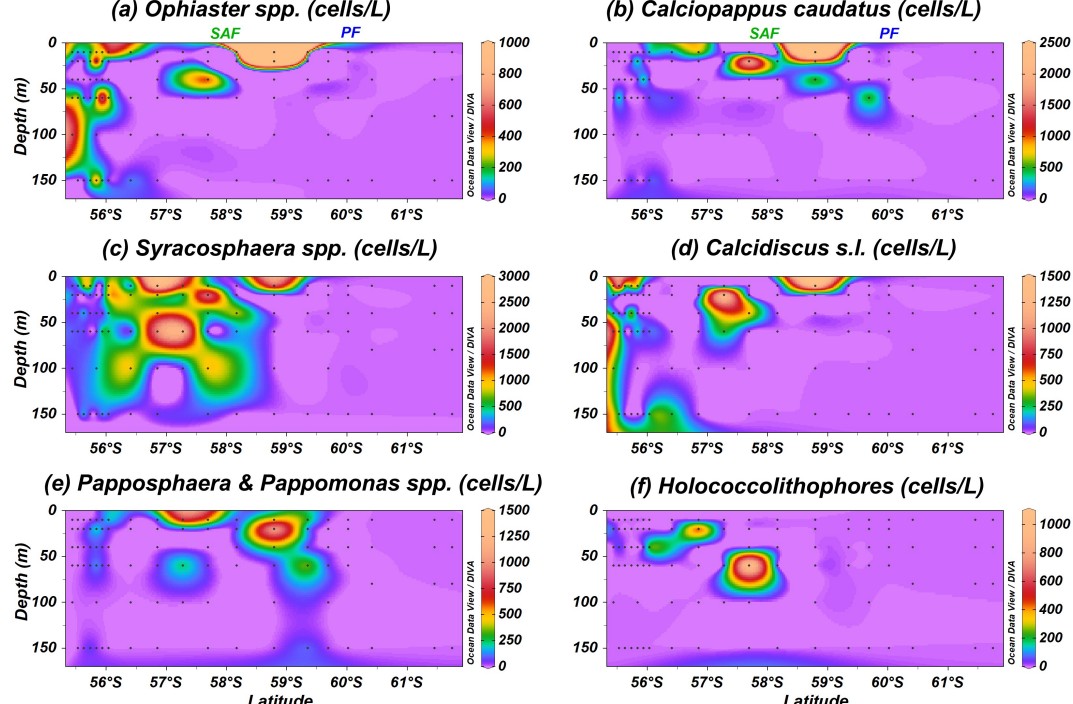

**Fig. 6. Location of the PS97 CTD stations studied showing the number of coccospheres/L of (a)** *Ophiaster* **spp., (b)** *Calciopappus caudatus*, **(c)** *Syracosphaera* **spp., (d)** *Calcidiscus* **s.l., (e)** *Papposphaera & Pappomonas* **spp., and (f) holococcolithophores. The Subantarctic Front (SAF) and the Polar Front (PF) according to Orsi et al. (1995) are indicated.**



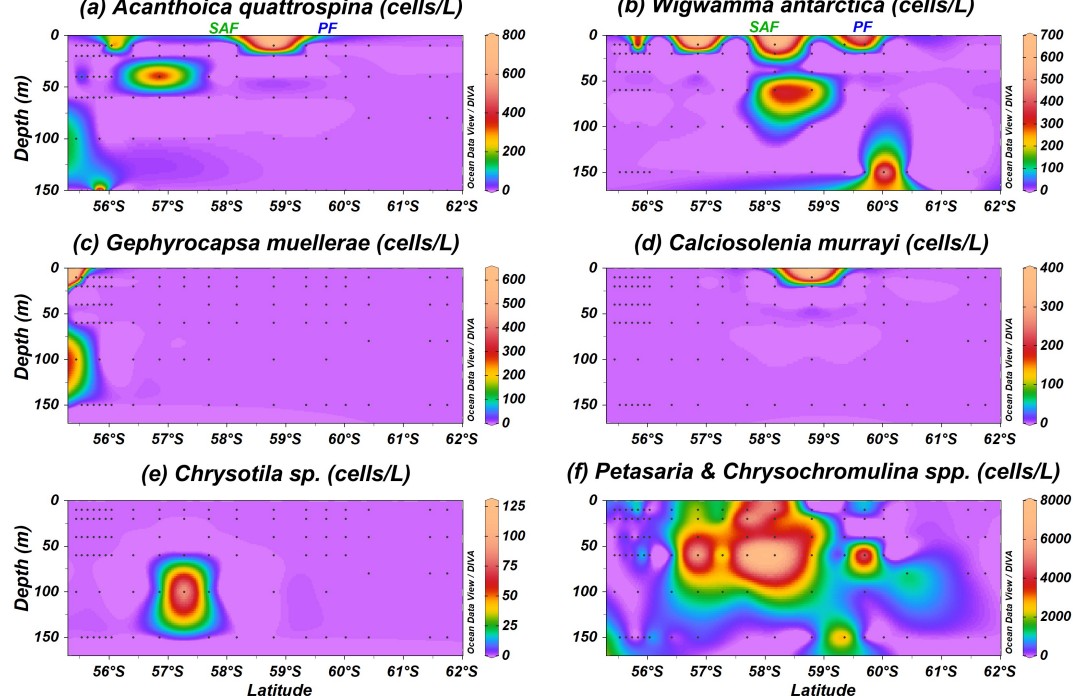

Fig. 7. Location of the PS97 CTD stations studied showing the number of coccospheres/L of (a) *Acanthoica quattrospina*, (b) *Wigwamma antarctica*, (c) *Gephyrocapsa muellerae*, (d) *Calciosolenia murrayi*, (e) *Chrysotila* sp., (d) *Petasaria* and *Chrysochromulina* spp. The Subantarctic Front (SAF) and the Polar Front (PF) according to Orsi et al. (1995) are indicated.





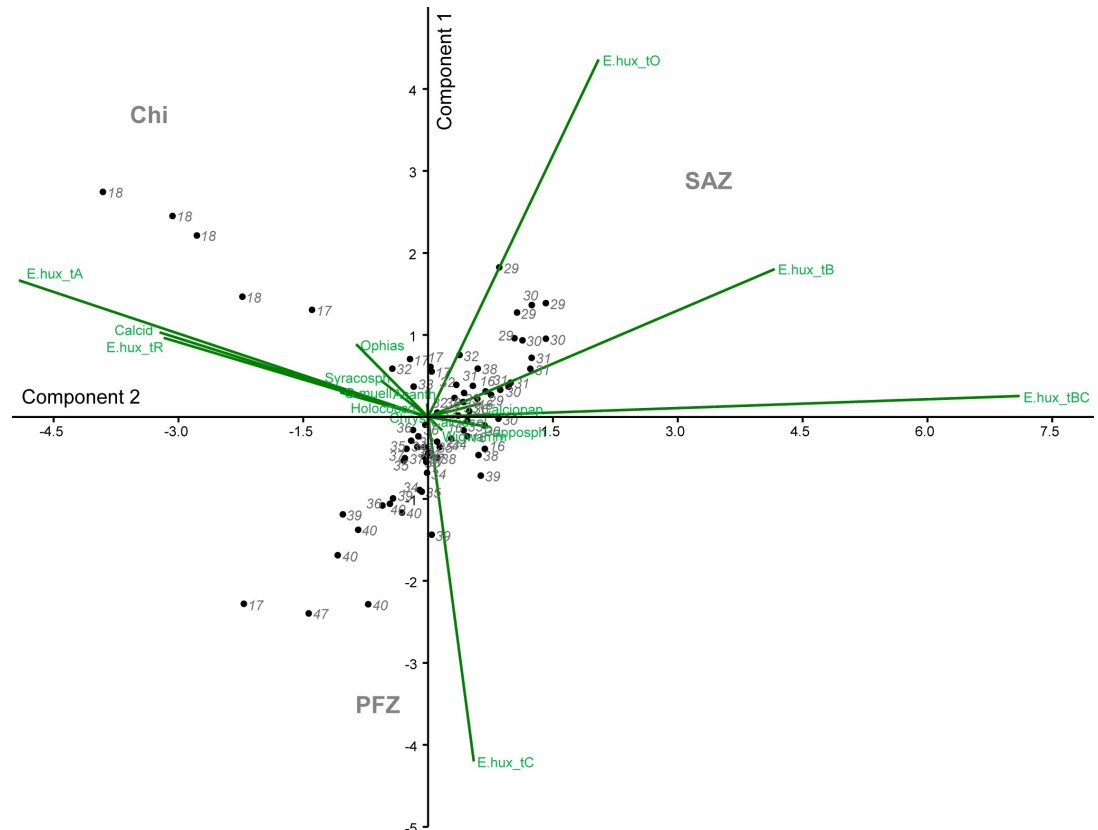

**Fig. 8. Principal Component Analysis (PCA) biplot performed on the coccolithophore dataset. The two main components, PC1 (y axis) and PC2 (x axis), coccolithophore taxa, station numbers and the different clusters are indicated. The following abreviations are used in the figure: E.hux_tA (*E. huxleyi* type A), E.hux_tB (*E. huxleyi* type B), E.hux_tBC (*E. huxleyi* type BC), E.hux_tC (*E. huxleyi* type C), E.hux_tO (*E. huxleyi* type O), E.hux_tR (*E. huxleyi* type R), Calcid (*Calcidiscus* s.l.), Acanth (*Acanthoica quattrospina*), Calciopap (*Calciopapus caudatus*), Calciosol (*Calciosolenia murrayi*), Chryso (*Chrysotila* sp.), G.mueller (*G. muellerae*), Holococo (Holococcolithophores), Ophias (*Ophiaster* spp.),Papposph (*Papposphaera* sp and *Pappomonas* spp.), Syracosph (*Syracosphaera* spp.), Wigwamm (*Wigwamma antarctica*), SAZ (subantarctic zone), PFZ (polar front zone) and Chi (Chilean coastal zone).**





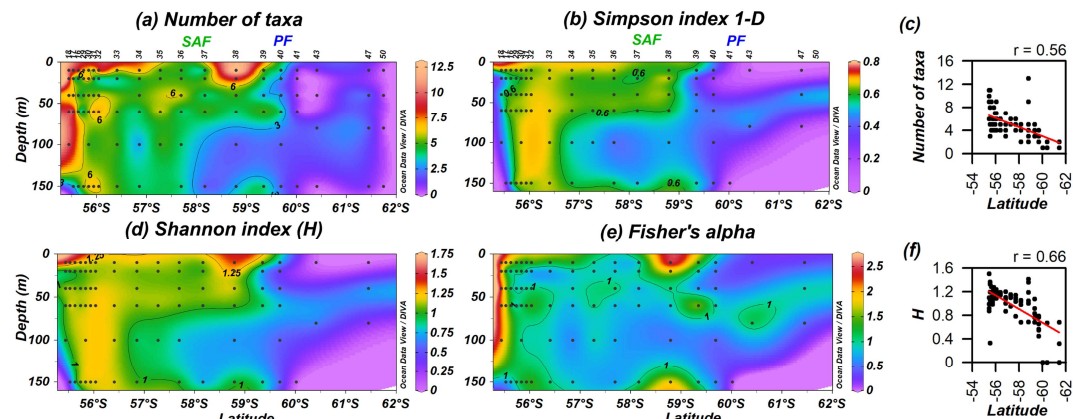

**Fig. 9. PS97 latitudinal transects showing coccolithophore diversity: (a) number of taxa, (b) Simpson index 1-D, (c) number of taxa vs latitude, (d) Shannon index (H), (e) Fisher's alpha and (f) H vs latitude. The Subantarctic Front (SAF) and the Polar Front (PF) according to Orsi et al. (1995) are indicated.**





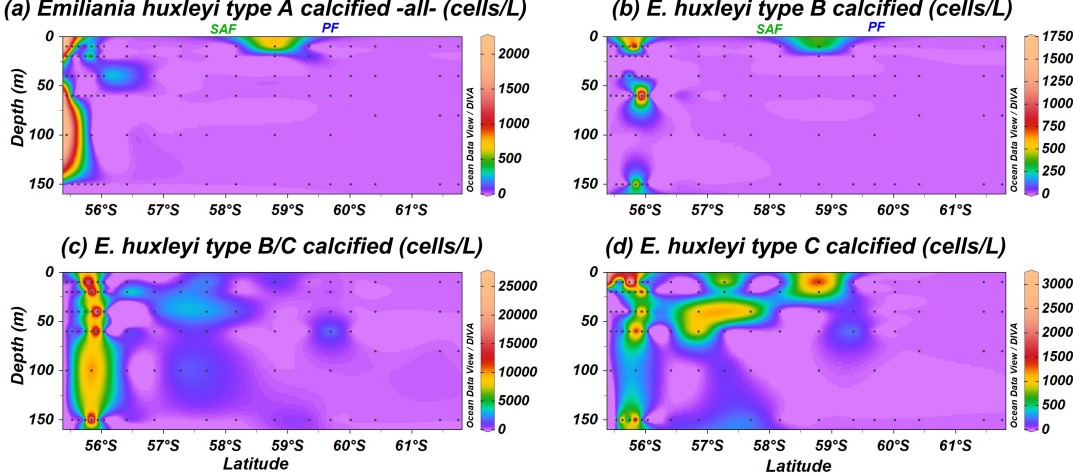

**Fig. 10.** Location of the PS97 CTD stations studied showing the number of coccospheres/L of *Emiliania huxleyi* calcified and overcalcified (a) type A group, (b) type B, (c) type B/C and (d) type C. The Subantarctic Front (SAF) and the Polar Front (PF) according to Orsi et al. (1995) are indicated.





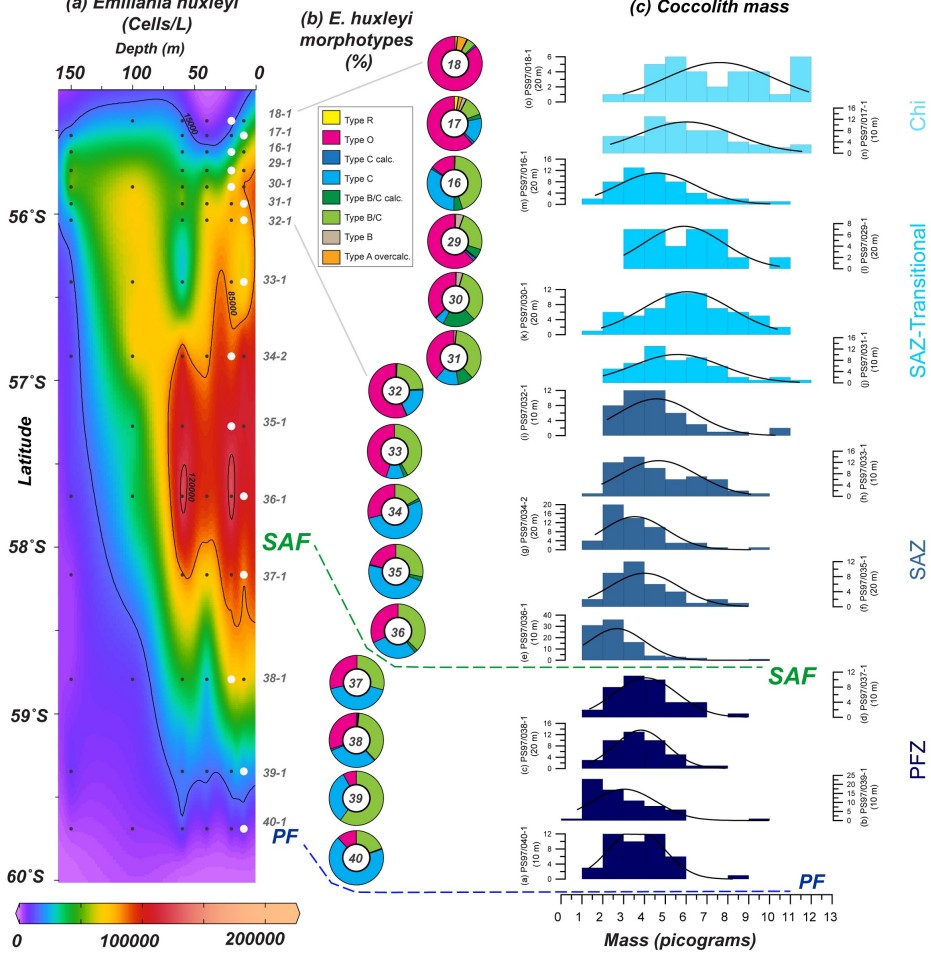

Fig. 11. (a) Location of the PS97 CTD stations between 55.4 and 60°S showing the number of cells/L of *Emiliania huxleyi* from 0 to
150 m. (b) Relative abundance of *E. huxleyi* morphotypes (including the different degree of calcification observed in SEM) in a
latitudinal transect (10-20 m, indicated with white dots in (a)), and (c) coccolith mass histograms at those specific locations. Bin size
in (c) is 1 picogram and normal distribution fitting line is shown. Station's number, different zones: Chi (Chilean coastal zone), SAZ-
transitional (from Chi to open ocean), SAZ (subantarctic zone), PFZ (polar front zone), and the main oceanographic fronts:
Subantarctic (SAF) and Polar Fronts (PF) according to Orsi et al. (1995) are indicated.





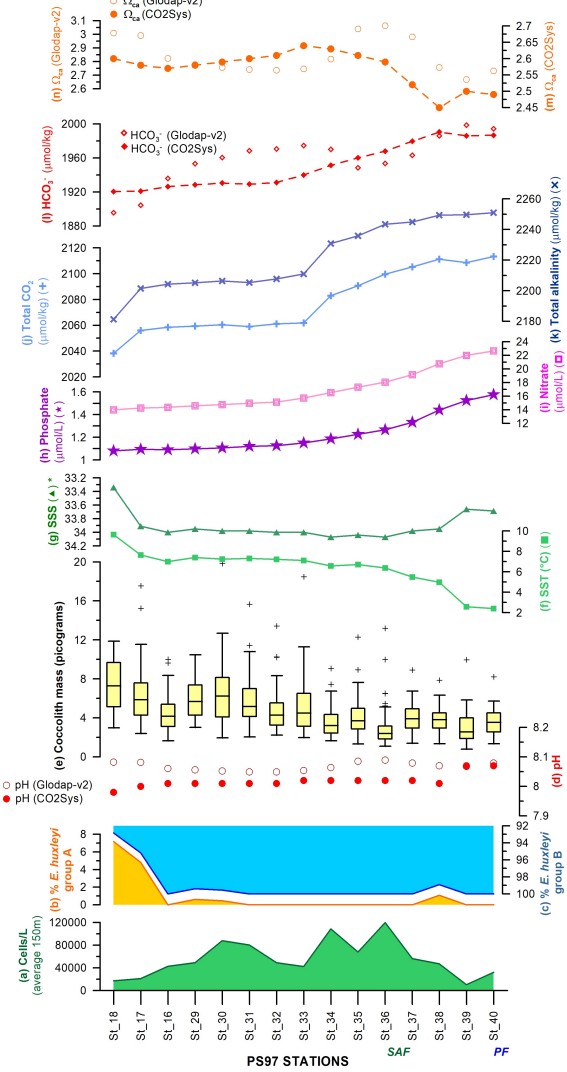

**Fig. 12. Latitudinal transect showing on the left hand side: (a) the uppermost 150 m average of cells/L, relative abundance of *E. huxleyi* morphotypes belonging to (b) group A and (c) to group B, (d) pH calculated with different approaches (Pierrot et al., 2006; Key et al., 2015), (e) coccolith mass boxplot (outliers have been indicated with "+"), (f) sea surface temperature (SST, °C) and (g)**
5 **salinity (* note the inverted scale) measured in situ (Lamy, 2016). Interpolated (h) phosphate (µmol/L) and (i) nitrate contents (µmol/kg) (Garcia et al., 2014), (j) total alkalinity (µmol/kg) and (k) total CO₂ (µmol/kg) (Goyet et al., 2000), (l) bicarbonate ion (HCO₃⁻, µmol/kg) and (m) calcite saturation state (Ω_Ca) calculated with different approaches (Pierrot et al., 2006; Key et al., 2015). The subantarctic (SAF) and Polar Fronts (PF) according to Orsi et al. (1995) are indicated.**



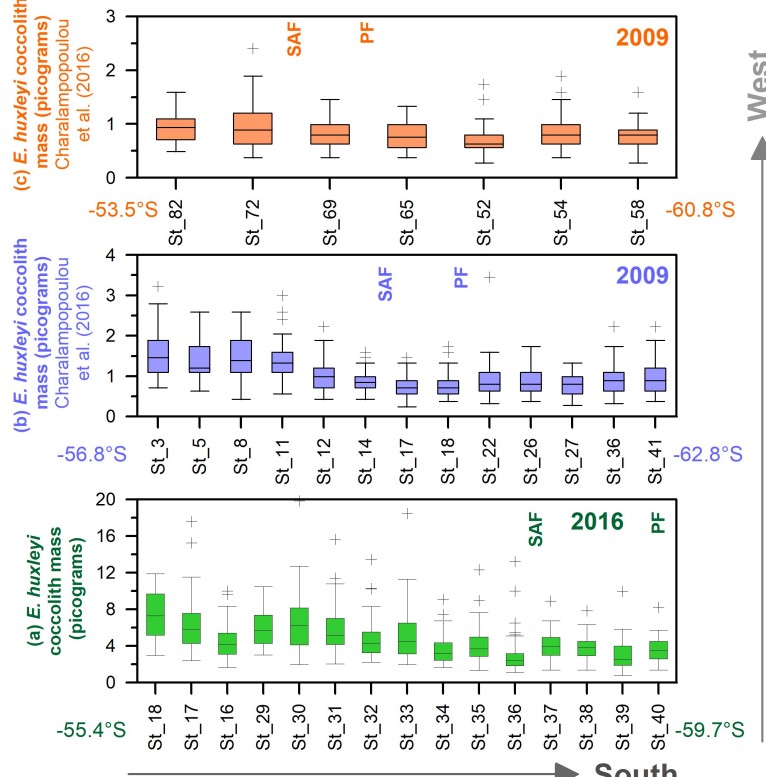

**Fig. 13.** Drake Passage latitudinal transects from East to West showing boxplot coccolith mass estimates (in pictograms): (a) this study, (b) transect at around 68°W Charalampopoulou et al., (2016), (c) transect at around 55-58°W Charalampopoulou et al., (2016). Note that (b, c) have been calculated from Charalampopoulou et al., (2016). Outliers have been indicated with "+" and numbers in the x axes refer to the original station numbers. The approximate location of the Subantarctic (SAF) and Polar Fronts (PF) are shown as well as the year of the sampling.



**Table 1. Classification scheme of *Emiliania huxleyi* morphotypes observed in the present study (modified from Hagino et al. (2011) regarding Young and Westbroek (1991), Cook et al. (2011) and Young et al. (2019)).**

| | *E. huxleyi* morphotype | Morphology of the distal shield | Morphology of the central area | Distal shield length | Comparable morphotypes in the literature |
|---|---|---|---|---|---|
| **A group** | Type A | Moderate-heavily calcified elements | Grill | <4 µm | Warm type (McIntyre and Bé, 1967), var. *huxleyi* (Medlin et al., 1996) |
| | Type A overcalcified | Moderate-heavily calcified elements | closed or nearly closed | <4 µm | Type A overcalcified (Young et al., 2003) |
| | Type R | *Reticulofenestra*-like heavily calcified distal shield elements | Grill | <4 µm | Type R (Young et al., 2003) |
| **B group** | Type B | Lightly calcified elements | solid plate | ≥4 µm | Type B (Young et al., 2003), var. *pujosae* (Verbeek, 1990) Medlin & Green in Medlin et al. (1996) |
| | Type B/C | Lightly calcified elements | solid plate | <4 µm | Type B/C (Young et al., 2003), var. *aurorae* (Cook, 2011) |
| | Type C | Lightly calcified elements | solid plate | <3.5µm | Cold type (McIntyre and Bé, 1967), Type C (Young et al., 2003), var. *kleijniae* (Medlin et al., 1996) |
| | Type O | Lightly calcified elements | opened or lamella | variable in size | Subarctic type (Okada and Honjo, 1973), Type B (Hagino et al., 2005) |

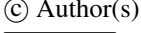



**Table 2. Summary of the coccolithophore taxa/groups in the studied plankton samples. The presence of each species in the Subantarctic Zone (SAZ), in the Polar Front Zone (PFZ) or in the Antarctic Zone (AZ) is indicated with "+" and with "−" if it is occasional. The station and water depth where the maximum number of cells/L is recorded has been also indicated.**

| Coccolithophore taxa / groups | SAZ | PFZ | AZ | Max. (cells/L) | Station | Depth (m) |
|---|---|---|---|---|---|---|
| *Acanthoica quattrospina* | + | - | | $0.7*10^3$ | PS97/038-1 | 10 |
| *Calcidiscus* s.l. | + | - | | $1.4*10^3$ | PS97/038-1 | 10 |
| *Chrysotila* sp. | - | | | $0.1*10^3$ | PS97/035-1 | 100 |
| *Calciosolenia murrayi* | + | - | | $0.3*10^3$ | PS97/038-1 | 10 |
| *Calcioppapus caudatus* | + | - | | $5.3*10^3$ | PS97/038-1 | 10 |
| *Emiliania huxleyi* type A (incl. overcalcified) | + | - | | $2.1*10^3$ | PS97/018-1 | 100 |
| *Emiliania huxleyi* type R | + | - | | $0.9*10^3$ | PS97/038-1 | 20 |
| *Emiliania huxleyi* type B | + | - | | $7.4*10^3$ | PS97/029-1 | 10 |
| *Emiliania huxleyi* type B/C | + | + | - | $74.0*10^3$ | PS97/030-1 | 150 |
| *Emiliania huxleyi* type C | + | + | - | $103.2*10^3$ | PS97/034-2 | 60 |
| *Emiliania huxleyi* Type O (incl. lamella) | + | + | - | $76.2*10^3$ | PS97/036-1 | 60 |
| *Gephyrocapsa muellerae* | + | | | $0.6*10^3$ | PS97/018-1 | 10 |
| Holococcolithophores | + | | | $1.0*10^3$ | PS97/036-1 | 60 |
| *Ophiaster* spp. (incl. *hydroideus* & *reductus*) | + | - | | $10.2*10^3$ | PS97/038-1 | 10 |
| *Papposphaera* sp. | - | | | $0.2*10^3$ | PS97/035-1 | 60 |
| *Pappomonas* spp. (incl. sp. 1 & 5) | + | + | | $1.4*10^3$ | PS97/038-1 | 20 |
| *Syracosphaera* spp. (incl. *dilatata, corolla, marginaporata, pulchra*) | + | - | | $2.8*10^3$ | PS97/034-2 | 10 |
| *Wigwamma antarctica* | - | + | - | $0.7*10^3$ | PS97/034-2 | 10 |





**Table 3. Correlation matrix between Principal component (PC) scores and the environmental variables. Significant Pearson correlation coefficients are indicated in bold (p<0.05)**

| Variables | PC 1 | PC 2 |
|---|---|---|
| SST | **0.77** | -0.01 |
| SSS | **-0.23** | **0.66** |
| Density | **-0.77** | **0.41** |
| Oxygen | **-0.70** | -0.02 |
| Fluorometer | -0.13 | -0.13 |
| Phosphate | **-0.70** | -0.21 |
| Nitrate | **-0.70** | -0.20 |
| Silicate | **-0.61** | -0.22 |




**Table 4. Correlation matrix between mean / median coccolith values and the environmental variables. Significant Pearson correlation coefficients are indicated in bold (p<0.05). Measurements in situ and calculated values are indicated.**

| | Environmental variables | Mean coccolith mass (pg) | Median coccolith mass (pg) |
|---|---|---|---|
| In situ | Temperature | **0.75** | **0.71** |
| | Salinity | -0.35 | -0.37 |
| | Fluorescence | 0.10 | 0.13 |
| Calculated | Phosphate | **-0.71** | **-0.66** |
| | Nitrate | **-0.75** | **-0.70** |
| | Silicate | -0.35 | -0.31 |
| | Total $CO_2$ | **-0.82** | **-0.77** |
| | Total Alkalinity | **-0.85** | **-0.81** |
| | $HCO_3^-$ Glodap-v2 | **-0.68** | **-0.66** |
| | HCO3- CO2Syt | **-0.81** | **-0.75** |
| | pH Glodap-v2 | -0.25 | -0.22 |
| | pH CO2Sys | **-0.70** | **-0.69** |
| | $CO_3^{2-}$ Glodap-v2 | 0.09 | 0.08 |
| | $CO_3^{2-}$ CO2Sys | 0.40 | 0.32 |
| | Ωca Glodap-v2 | 0.10 | 0.10 |
| | Ωca CO2Sys | 0.39 | 0.31 |



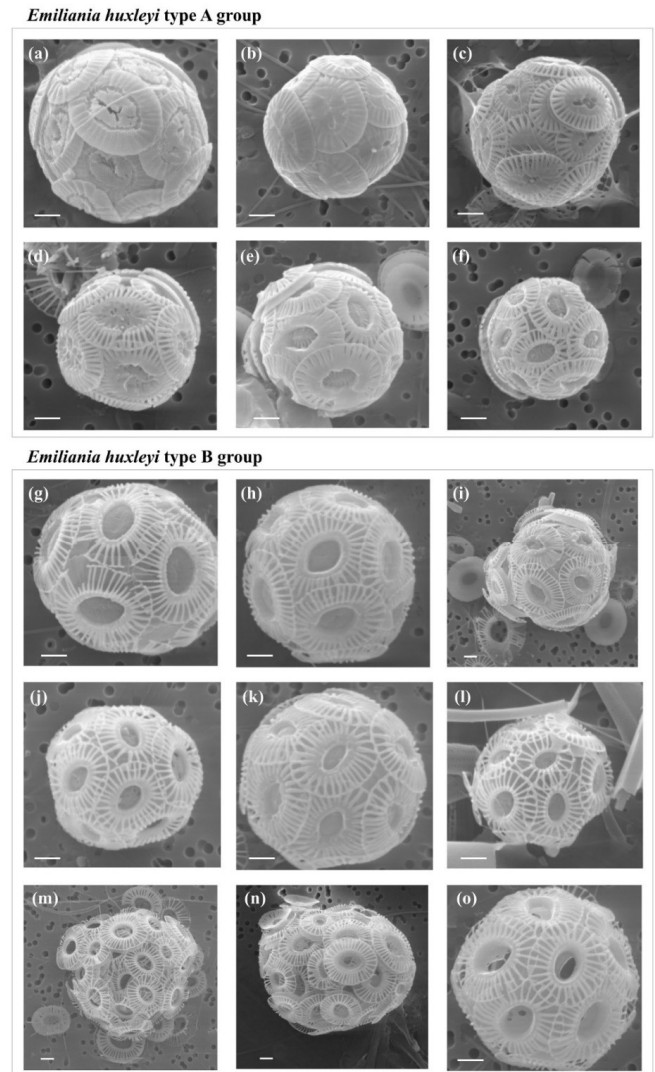

Plate 1. Specimens of *Emiliania huxleyi*. Group A includes (a) type R, sample PS97/018-1 at 10 m water depth; (b) type A overcalcified, PS97/018-1 at 60 m; (c) type A overcalcified, PS97/018-1 at 100 m; (d) type A calcified, PS97/018-1 at 60 m; (e) type A calcified PS97/018-1 at 20 m; (f) type A slightly calcified, PS97/018-1 at 60 m.





*Emiliania huxleyi* **group B includes: (g) type B PS97/016-1 at 5 m; (h) type B PS97/029-1 at 60 m; (i) type B calcified, PS97/29-1 at 150 m; (j) type B/C PS97/016-1 at 10 m; (k) type B/C slightly calcified PS97/030-1 at 150 m; (l) type C PS97/043-1, at 10 m; (m) type O -opened-, PS97/032-1 at 10 m; (n) type O -lamella-, PS97/038-1 at 10 m; (o) type O -lamella- PS97/017-1 at 60 m. The scale bar indicates 1 μm.**





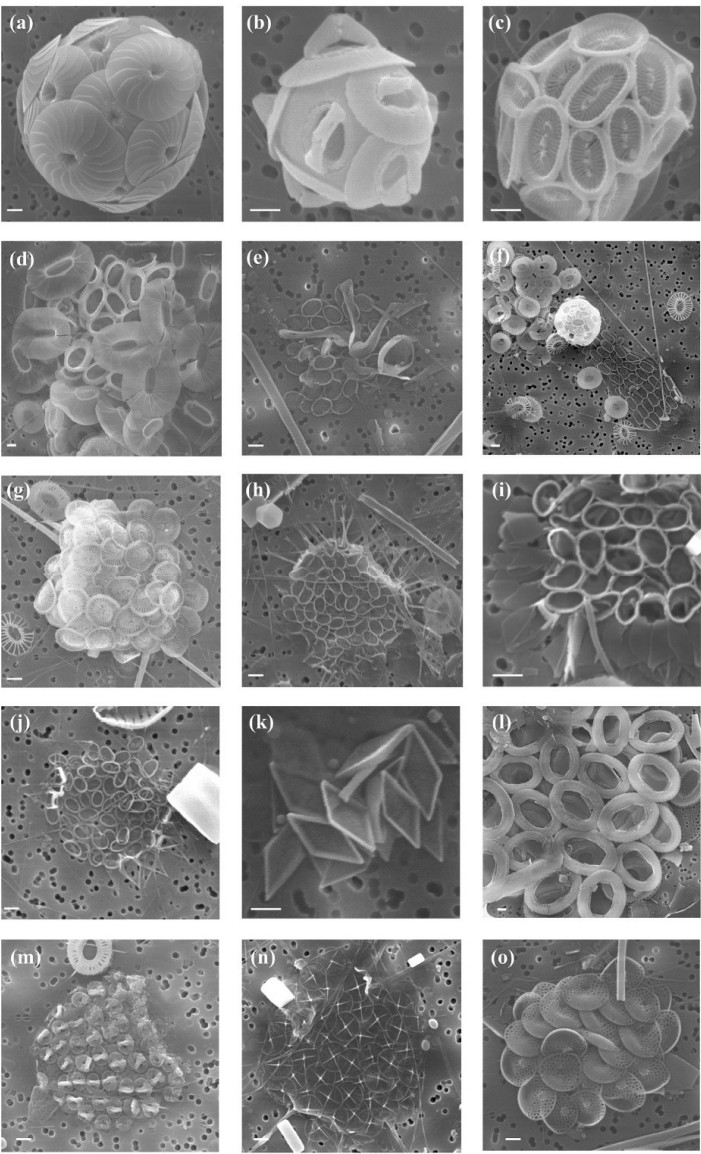

**Plate 2. Other taxa present in the study area.** Specimens of **(a)** *Calcidiscus leptoporus* s.s., sample PS97/016-1 at 5 m water depth; **(b)** *Gephyrocapsa muellerae*, PS97/018-1 at 10 m; **(c)** *Syracosphaera dilatata*, PS97/32-1 at 10 m; **(d)** *Syracosphaera corolla*, PS97/38-1 at 10 m; **(e)** *Ophiaster hydroideus* , PS97/38-1 at 10 m; **(f)** *Calciopappus caudatus* and *Emiliania huxleyi*, PS97/34-2 at 10 m; **(g)**



*Acanthoica quattrospina*, **PS97/32-1 at 10 m; (h)** *Papposphaera sagitiffera* **sp., PS97/38-1 at 20 m; (i)** *Papposphaera* **cf.** *heldalii*, **PS97/30-1 at 40 m; (j)** *Wigwamma antarctica,* **PS97/39-1 at 10 m; (k)** *Calciosolenia* **sp***.,* **PS97/38-1 at 10 m; (l)** *Chrysotila* **sp., PS97/35-1 at 100 m; (m)** *Syracosphaera strigilis* **HOL, PS97/17-1 at 40 m; (n)** *Chrysochromulina* **sp.; (o)** *Petasaria heterolepis* **sp., PS97/38-1 at 40 m. The scale bar indicates 1 µm.**

