# Peer review of "Calcification and latitudinal distribution of extant coccolithophores across the Drake Passage during late austral summer 2016"

_Biogeosciences, 2019_

## Referee Comment (RC1) · Anonymous Referee #1 · 12 Jun 2019

Summary:

Saavedra-Pellitero and co-authors present coccolithophore measurements from along the western side of the Drake Passage. They quantify coccolithophore species counts and coccolith mass. They present some oceanographic measurements but also use global databases to retrieve nutrients and carbonate chemistry quantities. They find a poleward decrease in both coccolithophore diversity and calcification. They group E. huxleyi morphotypes into two major groups (A and B) and are able to show that the southward decrease in calcification is related to a shift from A morphotypes along the Chilean margin to B morphotypes in the subantarctic and polar front zones. Coccolithophore calcification is inversely related to alkalinity, dissolved inorganic carbon, and pH. A Principle component analysis reveals three distinct clusters: Chilean coastal, SAZ, and PFZ. Temperature seems to be an important factor in controlling the distribution of coccolithophore species, as well as overall coccolithophore abundance and calcification.

General comments:

This manuscript is well written and of high quality. The authors present a valuable dataset with respect to observed coccolithophores. They present fantastic detailed plots of coccolithophore species in this Southern Ocean transect. This transect is slightly westward of the transects presented Charalampopoulou et al. (2016), which is a similar study. This manuscript offers more information on depth variations in coccolithophore abundances than previous studies in this region, which is great! This study reaches much of the same overall conclusions as previous transects observing coccolithophores in the Southern Ocean, so it is not groundbreaking, but adds to a solid overall conclusion of coccolithophores transitioning from more calcified species/morphotypes in the subtropics to less calcified ones in the ACC region. The conclusion that temperature is a controlling factor on coccolithophore abundance agrees with previous studies (e.g., Charalampopoulou et al., 2016). I think this manuscript is in great shape and only needs minor revisions. One piece that is missing is a bit more specific speculation about how coccolithophore abundance/calcification could change with climate change. The authors say that coccolithophores will be strongly influenced, but not how they will be influenced. I think it's important to hypothesize the direction of change, given current observations and relationships with environmental variables presented in the study. I also think that the positive relationship between temperature and coccolith mass needs to be emphasized a bit more. It is a bit of a shame that nutrients and carbonate chemistry parameters were not measured in situ, but I do not think that having these measurements would have changed the conclusions (it would have just added more strength to them). I also think that the depth

variations between the three different oceanic region clusters could be more emphasized (especially because this was not as well presented in previous studies, so I find it to be new information): maximum depth of coccolithophores decreases poleward.

Specific comments:

Page 1, Abstract: maybe add in something about the decreasing depth of coccolithophores as you go poleward (as shown in Figure 3a)

Page 2, Line 6: extra "substantial"

Page 2, Lines 11-13: This sentence is awkward and a bit hard to understand. Maybe it would be best rewritten like this: "Coccolithophores produce up to ∼40% of open ocean calcium carbonate (Poulton et al., 2013) and are responsible for ∼20% of global net marine primary production (Malone et al., 2017). Therefore, how coccolithophores respond to changing oceanic conditions is of upmost importance for marine ecology and carbon cycling."

Page 2, Line 32: I think that it's important to include that the Beaufort et al (2011) study includes both modern samples and paleodata from the last 40000 years. Maybe just add "over long timescales": "A known positive correlation exists over long timescales between surface-ocean…."

Page 3, Line 6: replace "actually" with "recently"

Page 3, Line 15: Perhaps replace "species levels" with "overall coccolithophore calcification" since Beaufort et al. (2011) and Freeman and Lovenduski (2015) both have drawn conclusions based on overall coccolithophore calcification. While the Beaufort study has some species level information, the Freeman and Lovenduski study does not.

Page 3, Line 23: Break this sentence up into two sentences for clarity: "Accordingly, we calculated extant coccolithophore species numbers at different stations between 10 and 150 m of the water column and evaluated the coccolith mass variations of

E. huxleyi. We compared these observations with in situ conductivity–temperature–depth (CTD) measurements, carbonate chemistry parameters, as well as to previously published Southern Ocean coccolithophore and calcification datasets."

Page 3, Line 28: no need to capitalize "stations"

Page 6, Line 7: instead of "a taxon" say "one taxon"

Page 7: Line 22: Add references to Figures 4 and 5: "..., grouped into A (Figure 4) and B (Figure 5) according to Young et al. (2019). Also, by "Young et al., 2019" do you mean Nanotax3 website? It is unclear what reference this is referring to in the bibliography.

Page 7, Line 30: Type A overcalcified and Type R seem very similar to me. How are they different exactly?

Page 8, Line 23: When you say that Syracosphaera dominates in the SAZ, do you mean that it dominates among the rare coccolithophore assemblage or among coccolithophores overall? Please modify to be more specific.

Page 9, Line 7: Take out the extraneous "the" before 77.4%.

Page 10, Lines 25-29: Could silica be becoming more limiting north of the PF, opening a niche for coccolithophores? Perhaps competition among phytoplankton is another possibility of coccolithophores increasing in abundance at the PF.

Page 11, Line 10: rather than saying "up to 61.7°S", maybe it's more appropriate to say "down to 61.7°S"

Page 11, Line 21: misspelling of the word "coastal"; and instead of saying "... increasing towards oceanic regions" maybe say "... increasing towards open ocean regions"

Page 12, Line 12: change "communities" to "community"

Page 12, Line 19: The widely used E. huxleyi strain NZEH (morphotype R) and E.

huxleyi strain RCC1216 (morphotype R) were both isolated from around New Zealand so I believe that would count as "observing" it there too. For example, see Methods in Iglesias-Rodriguez et al. (2017) and Langer et al., (2009):

Iglesias-Rodriguez, Maria Debora, Bethan M. Jones, Sonia Blanco-Ameijeiras, Mervyn Greaves, Maria Huete-Ortega, and Mario Lebrato. "Physiological responses of coccolithophores to abrupt exposure of naturally low pH deep seawater." PloS one 12, no. 7 (2017): e0181713.

Langer, Gerald, Gernot Nehrke, Ian Probert, J. Ly, and Patrizia Ziveri. "Strain-specific responses of Emiliania huxleyi to changing seawater carbonate chemistry." Biogeosciences 6, no. 11 (2009): 2637-2646.

Page 12, Line 30: change "up to ca. 6°C" to "down to ca. 6°C"

Page 13, Line 20: change "decreases" to "decrease"

Page 14, section 4.3 in general: I think that the temperature as a controlling factor needs to be discussed more. It's in the abstract (Page 1, Line 28/29) as a greater limiting factor than carbonate chemistry, which I totally agree with, but I think it needs more discussion in the paper. Are colder temperatures in the poleward direction selecting for lightly calcified species/morphotypes? Or could it be a physiological change induced by colder temperatures? I think a bit more speculation (perhaps bringing in some laboratory experiments) would be nice in this section. There's a summary of the effects of temperature on coccolithophore calcification in Krumhardt et al. (2017): Krumhardt, Kristen M., Nicole S. Lovenduski, M. Debora Iglesias-Rodriguez, and Joan A. Kleypas. "Coccolithophore growth and calcification in a changing ocean." Progress in oceanography 159 (2017): 276-295.

Page 14, Line 20: misspelled the word "mass"

Page 14, Lines 20 – 25: It's good that you pointed out the fact that the carbonate chemistry parameters have been estimated, rather than measured. However, the latitudinal

gradients in carbonate chemistry parameters are pretty well established and I don't think it would affect the relationships you're seeing.

Page 15, Line 3: misspelled the word "Striking"

Page 15, Line 10. Here is where it would be good to speculate on the direction of change in coccolithophore abundance/calcification (or latitudinal species/morphotype shifts) with ongoing climate change. You could bring up the positive correlation with temperature shown in Table 4 and the PC analysis.

Figure 13: I like that you included this comparison to the Charalampopoulou et al. (2016) paper. Is the direction arrow on the right hand side of the figure supposed to say "East" rather than "West"?. I thought the Charalampopoulou transects were to the east of the present study...

Table 1 and Plate 1: I like that you grouped the E hux morphotypes into 2 main groups. There seems to be a fluidity between all these morphotypes and grouping into only 2 groups makes the information much more digestible.

Table 4: misspelling on the line HCO3- CO2Sys

---

## Referee Comment (RC2) · Anonymous Referee #2 · 18 Jun 2019

The paper presents well documented distributional data of coccolithophores and in particular E. huxleyi morphotypes across the oceanographic fronts in the area of the Drake Passage, an important zone for monitoring the path of the Antarctic Circumpolar Current. The presentation of the methods and data is clear and the discussioni is well supported, showing a consistent latitudinal trend of decreasing coccolith mass along with temperature decrease and a gradient in carbonate chemistry parameters. Overall the manuscript represent a substantial contribution in the field of coccolithophore studies, adding new information and providing accurate measurements of both Ehux types and coccolith mass. The data are well presented, with figures and plates are of excellent quality. Some points deserve further discussion, in particular: section 4.3

is well organized and points to significant changes in the coccolith mass of E. huxleyi across the different fronts and zones of the ACC. However, while the degre of calcification is considered as the main driver of coccolith mass variation (but the assessment of degree of calcification and mass is done with different techniques, so there can be no direct attribution), there is no discussion about the influence of coccolith size on cocolith mass, e.g. type C is smaller than B/C which is smaller than type B, by definition. Carbonate chemistry parameters. The discussion of the relation between coccolithophore calcification and the carbonate chemistry of the water column should be considered even more carefully, given the fact that the data are not measured in the same samples and the pattern of pH variation is not so clear - a different trend appears if the different calculations are considered, e.g. fig. 12. However, the correlation with  $\Omega$ calcite seems meaningful, looking at the graphs, but this parameter is not considered in the discussion. The last sentence of the conclusion is however not supported and does not explain how climate change will affect the calcification mode of coccolithophores. given that no clear relationship between the degree of calcification and the carbonate chemistry of sea water are established yet, but rather different correlations seem to exist in different areas of the world oceans and under different oceanographic conditions, so the question remains open. Page 11 line 12: also Malinverno et al., 2016 show the shift in dominance from coccolithophores to diatoms in water samples across the PF / sACCf, so this could be cited.

Typos: Page 1, line 22: classified » identified Page 2, line 6: delete "substantial" which is repeated twice Page 2, line 7: dissolved carbon » dissolved inorganic carbon Page 2, line 16: phosphate is mis-spelled Page 2, line 28: the future » in the future Page 4, line 6: (2004) is repeated Page 7 line 20: this taxa » this taxon Page 8 line 1: later » latter Page 13 line 5: established » established by

---

## Short Comment (SC1) · 21 Jun 2019

Dear Saavedra-Pellitero and co-authors, I have read your paper in BGS discussion, and I have a short comment to make: the description of the central area of the heavy-calcified morphotype-R as "Grill" is not aligned with recently observation in the Eastern South Pacific, in which, it's reported as completely or nearly completely covered (see Fig. 3b in Beaufort et al., 2011 and Fig. 2d in von Dassow et al., 2018).

---

## Author Response (AR2)

**Fachbereich 5 – Geowissenschaften**

[Figure]

**Fachgebiet Paläozeanographie /Sedimentologie**

**Dr. Mariem Saavedra-Pellitero**

FB Geowissenschaften
Klagenfurter Straße
Postfach 33 04 40
D - 28334 Bremen

Telefon     (49) 0421 / 218-99501
Telefax     (49) 0421 / 218-65219
E-mail      msaavedr@uni-bremen.de

Bremen, August 16th 2019

Dear Editor(s),

Please find attached the revised manuscript "**Calcification and latitudinal distribution of extant coccolithophores across the Drake Passage during late austral summer 2016"** by Mariem Saavedra-Pellitero* (corresponding author), Karl-Heinz Baumann, Miguel Angel Fuertes, Hartmut Schulz, Yann Marcon, Nele Manon Vollmar, Jose Abel Flores and Frank Lamy.

We addressed all the comments and suggestions by the two referees and other members of the scientific community.

Additionally, we found a mistake in Figure 2. The units in (d) dissolved oxygen should be in µmol/kg, instead of mL/L. Therefore we changed it in the figure and in the figure caption. These are the only two changes we did since the last version submitted (13th of August, 2019)

All the best,

Mariem Saavedra-Pellitero

**Fachbereich 5 – Geowissenschaften**

**Fachgebiet Paläozeanographie /Sedimentologie**

**Dr. Mariem Saavedra-Pellitero**

[Figure]

FB Geowissenschaften
Klagenfurter Straße
Postfach 33 04 40
D - 28334 Bremen

Telefon      (49) 0421 / 218-99501
Telefax      (49) 0421 / 218-65219
E-mail       msaavedr@uni-bremen.de

Bremen, August 7th 2019

Dear Editor(s),

Please find attached the revised manuscript initially entitled **"Calcification and distribution of extant coccolithophores across the Drake Passage during late austral summer 2016"** by Mariem Saavedra-Pellitero* (corresponding author), Karl-Heinz Baumann, Miguel Angel Fuertes, Hartmut Schulz, Yann Marcon, Nele Manon Vollmar, Jose Abel Flores and Frank Lamy.

We addressed all the comments and suggestions by the two referees and other members of the scientific community and we thank you for the 3 weeks extension you gave us.

If possible, we would like to take the chance to change the title to **"Calcification and latitudinal distribution of extant coccolithophores across the Drake Passage during late austral summer 2016"**, because we think it describes better this piece of research.

All the best,

Mariem Saavedra-Pellitero

**Comments:**

We thank all the reviewers for their comments and suggestions which definitely helped us to improve this manuscript.
The reply to the reviewers is structured as follows:
Comments from the reviewers in black
Our reply to reviewers in red.
Note that line numbers refer to the 1st submitted version, not the revised one.

**Francisco Díaz-Rosas**

Dear Saavedra-Pellitero and co-authors, I have read your paper in BGS discussion, and I have a short comment to make: the description of the central area of the heavy calcified morphotype-R as "Grill" is not aligned with recently observation in the Eastern South Pacific, in which, it's reported as completely or nearly completely covered (see Fig. 3b in Beaufort et al., 2011 and Fig. 2d in von Dassow et al., 2018).

We thank Francisco Rias-Rosas for pointing out this. We absolutely agree with him, and we accordingly modified Table 1. The morphology of the central area is now described as "covered or nearly covered" and we also added the reference "von Dassow et al. (2018)" in the last column (comparable morphotypes in the literature)

**Anonymous Referee #1**

**Summary:**
Saavedra-Pellitero and co-authors present coccolithophore measurements from along the western side of the Drake Passage. They quantify coccolithophore species counts and coccolith mass. They present some oceanographic measurements but also use global databases to retrieve nutrients and carbonate chemistry quantities. They find a poleward decrease in both coccolithophore diversity and calcification. They group E. huxleyi morphotypes into two major groups (A and B) and are able to show that the southward decrease in calcification is related to a shift from A morphotypes along the Chilean margin to B morphotypes in the subantarctic and polar front zones. Coccolithophore calcification is inversely related to alkalinity, dissolved inorganic carbon, and pH. A Principle component analysis reveals three distinct clusters: Chilean coastal, SAZ, and PFZ. Temperature seems to be an important factor in controlling the distribution of coccolithophore species, as well as overall coccolithophore abundance and calcification.

**General comments:**
Referee #1 (R#1): This manuscript is well written and of high quality. The authors present a valuable dataset with respect to observed coccolithophores. They present fantastic detailed plots of coccolithophore species in this Southern Ocean transect. This transect is slightly westward of

the transects presented Charalampopoulou et al. (2016), which is a similar study. This manuscript offers more information on depth variations in coccolithophore abundances than previous studies in this region, which is great! This study reaches much of the same overall conclusions as previous transects observing coccolithophores in the Southern Ocean, so it is not groundbreaking, but adds to a solid overall conclusion of coccolithophores transitioning from more calcified species/morphotypes in the subtropics to less calcified ones in the ACC region. The conclusion that temperature is a controlling factor on coccolithophore abundance agrees with previous studies (e.g., Charalampopoulou et al., 2016). I think this manuscript is in great shape and only needs minor revisions.

Mariem Saavedra-Pellitero et al. (MSP): We thank reviewer #1 for his/her insightful comments and we agree with most of them and made changes accordingly.

R#1: One piece that is missing is a bit more specific speculation about how coccolithophore abundance/calcification could change with climate change. The authors say that coccolithophores will be strongly influenced, but not how they will be influenced. I think it's important to hypothesize the direction of change, given current observations and relationships with environmental variables presented in the study. I also think that the positive relationship between temperature and coccolith mass needs to be emphasized a bit more.
MSP: This point has been made also by reviewer #2.
We included more information regarding the missing information at the end of section 4.3.

R#1: It is a bit of a shame that nutrients and carbonate chemistry parameters were not measured in situ, but I do not think that having these measurements would have changed the conclusions (it would have just added more strength to them).
MSP: We absolutely agree with reviewer#1 and we are aware that it is a limitation we have face in this study.

R#1: I also think that the depth variations between the three different oceanic region clusters could be more emphasized (especially because this was not as well presented in previous studies, so I find it to be new information): maximum depth of coccolithophores decreases poleward.
MSP: We added more details regarding depth variations for each of the clusters/zones in section 4.2.
Additionally, we mentioned it now in the abstract, conclusions and also briefly in section 4.1.

**Specific comments:**
R#1: Page 1, Abstract: maybe add in something about the decreasing depth of coccolithophores as you go poleward (as shown in Figure 3a)
MSP: We wrote: "We find that coccolithophore abundance, diversity and maximum depth habitat decrease southwards marking different oceanographic fronts as ecological boundaries" in the abstract.

R#1: Page 2, Line 6: extra "substantial".
MSP: The extra "substantial" was deleted.

R#1: Page 2, Lines 11-13: This sentence is awkward and a bit hard to understand. Maybe it would be best rewritten like this: "Coccolithophores produce up to _40% of open ocean calcium carbonate (Poulton et al., 2013) and are responsible for _20% of global net marine primary production (Malone et al., 2017). Therefore, how coccolithophores respond to changing oceanic conditions is of upmost importance for marine ecology and carbon cycling."
MSP: We rewrote this sentence (literally) using the suggestion from reviewer #1.

R#1: Page 2, Line 32: I think that it's important to include that the Beaufort et al (2011) study includes both modern samples and paleodata from the last 40000 years. Maybe just add "over long timescales": "A known positive correlation exists over long timescales between surface-ocean: : :."
MSP: We agree and therefore we added "over long timescales" to the text.

R#1: Page 3, Line 6: replace "actually" with "recently"
MSP: We replaced "actually" with "recently".

R#1: Page 3, Line 15: Perhaps replace "species levels" with "overall coccolithophore calcification" since Beaufort et al. (2011) and Freeman and Lovenduski (2015) both have drawn conclusions based on overall coccolithophore calcification. While the Beaufort study has some species level information, the Freeman and Lovenduski study does not.
MSP: Following the suggestion of reviewer#1 we made this sentence simpler, and wrote: "Even with a temperature-driven range expansion of coccolithophores in the SO, surface ocean carbonate chemistry is now capable of exerting a first-order control on the composition of coccolithophore assemblages as well as on overall coccolithophore calcification (Cubillos et al., 2007; Mohan et al., 2008; Beaufort et al., 2011; Freeman and Lovenduski, 2015)".

R#1: Page 3, Line 23: Break this sentence up into two sentences for clarity: "Accordingly, we calculated extant coccolithophore species numbers at different stations between 10 and 150 m of the water column and evaluated the coccolith mass variations of E. huxleyi. We compared these observations with in situ conductivity–temperature– depth (CTD) measurements, carbonate chemistry parameters, as well as to previously published Southern Ocean coccolithophore and calcification datasets."
MSP: We spit the sentence into two shorter ones.

R#1: Page 3, Line 28: no need to capitalize "stations"
MSP: We changed it to "stations".

R#1: Page 6, Line 7: instead of "a taxon" say "one taxon"
MSP: We use "one" instead of "a".

R#1: Page 7: Line 22: Add references to Figures 4 and 5: ": : :, grouped into A (Figure 4) and B (Figure 5) according to Young et al. (2019). Also, by "Young et al., 2019" do you mean Nanotax3 website? It is unclear what reference this is referring to in the bibliography.
MSP: We added the references to Figures 4 and 5 as suggested.

Yes, as pointed out by reviewer#1, we unclearly referenced Nanotax3 website in the previous version. To ensure that we cite it correctly this time, we double checked in http://www.mikrotax.org/Nannotax3/pages/ntax-citation.html and referenced accordingly.

R#1: Page 7, Line 30: Type A overcalcified and Type R seem very similar to me. How are they different exactly?
MSP: They are indeed similar and showed similar distribution, but they still show slightly different morphologies, as shown in Plate 1.
In *E. huxleyi* type R the slits between distal shield elements are almost or totally closed, and the tube is usually thick (Plate 1 a) giving a *Reticulofenestra*-like appearance, while type A overcalcified shows just a closed or nearly closed central area (Plate 1 b, c), but not almost closed slits.
Since this information can be found in the original sources mentioned in the section 2.1, we simply added the references to the specific pictures of the different morphotypes displayed Plate 1, e.g., "Type R (Plate 1 a)…"

R#1: Page 8, Line 23: When you say that Syracosphaera dominates in the SAZ, do you mean that it dominates among the rare coccolithophore assemblage or among coccolithophores overall? Please modify to be more specific.
MSP: We specify now that it dominates among the rare coccolithophore assemblage.

R#1: Page 9, Line 7: Take out the extraneous "the" before 77.4%.
MSP: "The" was deleted.

R#1: Page 10, Lines 25-29: Could silica be becoming more limiting north of the PF, opening a niche for coccolithophores? Perhaps competition among phytoplankton is another possibility of coccolithophores increasing in abundance at the PF.
MSP: This is a likely possibility, as far as we know from other papers in the Atlantic sector of the Southern Ocean (e.g., Smith et al., 2017). However, we do not have in situ silica measurements or diatoms counts, so we decided to avoid speculation and did not to include this suggestion in the new version of the manuscript.
Future research could usefully address the interesting shift in dominance from coccolithophores to diatoms across the PF (also mentioned by reviewer #2) and assess the interrelationship between both groups, even at a sub-species level.

R#1: Page 11, Line 10: rather than saying "up to 61.7_S", maybe it's more appropriate to say "down to 61.7_S"
MSP: We changed it to "down to".

R#1: Page 11, Line 21: misspelling of the word "coastal"; and instead of saying ": : : increasing towards oceanic regions" maybe say ": : : increasing towards open ocean regions"
MSP: We made both changes

Page 12, Line 12: change "communities" to "community"

MSP: We could not find the word "communities" in page 12 line 12, so we assumed that reviewer#1 was referring to page 12 line 3. We changed "communities" to "community" in that sentence.

R#1: Page 12, Line 19: The widely used E. huxleyi strain NZEH (morphotype R) and E. huxleyi strain RCC1216 (morphotype R) were both isolated from around New Zealand so I believe that would count as "observing" it there too.
For example, see Methods in Iglesias-Rodriguez et al. (2017) and Langer et al., (2009):
Iglesias-Rodriguez, Maria Debora, Bethan M. Jones, Sonia Blanco-Ameijeiras, Mervyn Greaves, Maria Huete-Ortega, and Mario Lebrato. "Physiological responses of coccolithophores to abrupt exposure of naturally low pH deep seawater." PloS one 12, no. 7
(2017): e0181713.
Langer, Gerald, Gernot Nehrke, Ian Probert, J. Ly, and Patrizia Ziveri. "Strain-specific responses of Emiliania huxleyi to changing seawater carbonate chemistry." Biogeosciences 6, no. 11
(2009): 2637-2646.
MSP: We thank reviewer#1 for those interesting papers. They are cited now in the new version.

R#1: Page 12, Line 30: change "up to ca. 6_C" to "down to ca. 6_C"
MSP: We changed it to "down to ca. 6_C".

R#1: Page 13, Line 20: change "decreases" to "decrease"
MSP: We changed it to "decrease".

R#1: Page 14, section 4.3 in general: I think that the temperature as a controlling factor needs to be discussed more. It's in the abstract (Page 1, Line 28/29) as a greater limiting factor than carbonate chemistry, which I totally agree with, but I think it needs more discussion in the paper.
MSP: We included more information regarding this point at the end of section 4.3.

R#1: Are colder temperatures in the poleward direction selecting for lightly calcified species/morphotypes? Or could it be a physiological change induced by colder temperatures?
MSP: These are in fact very interesting questions that we will keep in mind for future work. However, we believe that we would need more data in order to properly answer them.
In the new version of the manuscript, we highlighted the relevant role of temperature, and mentioned the degree of adaptive potential of coccolithophores (last sentence in section 4.3), but we did not want to speculate more.

R#1: I think a bit more speculation (perhaps bringing in some laboratory experiments) would be nice in this section.
MSP: We added more speculation (and more references citing culture experiments) in section 4.3.

R#1: There's a summary of the effects of temperature on coccolithophore calcification in Krumhardt et al. (2017): Krumhardt, Kristen M., Nicole S. Lovenduski, M. Debora Iglesias-Rodriguez, and Joan A. Kleypas. "Coccolithophore growth and calcification in a changing ocean." Progress in oceanography 159 (2017): 276-295.

MSP: We thank reviewer#1 for suggesting this paper. We cited it in the new version in section 4.3.

R#1: Page 14, Line 20: misspelled the word "mass"
MSP: We corrected it.

R#1: Page 14, Lines 20 – 25: It's good that you pointed out the fact that the carbonate chemistry parameters have been estimated, rather than measured. However, the latitudinal gradients in carbonate chemistry parameters are pretty well established and I don't think it would affect the relationships you're seeing.
MSP: We thank reviewer#1 for this positive comment.

R#1: Page 15, Line 3: misspelled the word "Striking"
MSP: We corrected it.

R#1: Page 15, Line 10. Here is where it would be good to speculate on the direction of change in coccolithophore abundance/calcification (or latitudinal species/morphotype shifts) with ongoing climate change. You could bring up the positive correlation with temperature shown in Table 4 and the PC analysis.
MSP: We speculated about possible future scenarios in section 4.3.

R#1: Figure 13: I like that you included this comparison to the Charalampopoulou et al. (2016) paper. Is the direction arrow on the right hand side of the figure supposed to say "East" rather than "West"?. I thought the Charalampopoulou transects were to the east of the present study: : :
MSP: We thank reviewer#1 for spotting this mistake. We modified Figure 13 and wrote "East", because those transects are east of our study.

R#1: Table 1 and Plate 1: I like that you grouped the E hux morphotypes into 2 main groups. There seems to be a fluidity between all these morphotypes and grouping into only 2 groups makes the information much more digestible.
MSP: We are glad that reviewer#1 appreciate our approach.

R#1: Table 4: misspelling on the line HCO3- CO2Sys
MSP: We corrected it.

**Anonymous Referee #2**

Referee #2 (R#2): The paper presents well documented distributional data of coccolithophores and in particular E. huxleyi morphotypes across the oceanographic fronts in the area of the Drake Passage, an important zone for monitoring the path of the Antarctic Circumpolar Current. The presentation of the methods and data is clear and the discussioni is well supported, showing a consistent latitudinal trend of decreasing coccolith mass along with temperature decrease and a gradient in carbonate chemistry parameters. Overall the manuscript represent a substantial

contribution in the field of coccolithophore studies, adding new information and providing accurate measurements of both Ehux types and coccolith mass.
The data are well presented, with figures and plates are of excellent quality.

Mariem Saavedra-Pellitero et al. (MSP): We thank reviewer#2 for his/her positive feedback and insightful comments. We made modifications in the text based on his/her suggestions.

R#2: Some points deserve further discussion, in particular: section 4.3 is well organized and points to significant changes in the coccolith mass of E. huxleyi across the different fronts and zones of the ACC.
However, while the degre of calcification is considered as the main driver of coccolith mass variation (but the assessment of degree of calcification and mass is done with different techniques, so there can be no direct attribution), there is no discussion about the influence of coccolith size on cocolith mass, e.g. type C is smaller than B/C which is smaller than type B, by definition.
MSP: This was one of the main challenges of this study: to choose the best way to characterize the coccolithophore assemblage while minimizing the error in the coccolith mass estimates. For this reason we choose to combine SEM analyses and C-calcita LM measurements.
Using just LM techniques would not have allowed us to distinguish the different *E. huxleyi* morphotypes (i.e., O type from B/C), and using only SEM techniques, would have implied to calculate the coccolith mass with the equations of Young and Ziveri (2000), being forced to assume some of the values (e.g., the shape-dependent constant $K_s$). That is the reason behind using different techniques for this research.
Regarding the influence of coccolith size on coccolith mass, we are aware that (by definition) there is size variation (Table 1) and certain overlap in size among different *E. huxleyi* morphotypes, which makes a direct comparison complicated. That is why we provided (as a first approach) Figure 11, but we decided to stick just to the mass variations in the manuscript. Biometric work would be required for the discussion that reviewer#2 is asking here, but we think that it is a topic for future research.

R#2: Carbonate chemistry parameters.
The discussion of the relation between coccolithophore calcification and the carbonate chemistry of the water column should be considered even more carefully, given the fact that the data are not measured in the same samples and the pattern of pH variation is not so clear – a different trend appears if the different calculations are considered, e.g. fig. 12.
MSP: Yes, we agree with reviewer#2, but as pointed out by reviewer#1, we made it very clear throughout the manuscript that those are not in situ measurements.
We believe that it has been well specified, and it is depicted even in the conclusions.

R#2: However, the correlation with omega calcite seems meaningful, looking at the graphs, but this parameter is not considered in the discussion.
MSP: We added more discussion in the section 4.3 regarding the weak correlation between calcite saturation and coccolith mass.

R#2: The last sentence of the conclusion is however not supported and does not explain how climate change will affect the calcification mode of coccolithophores, given that no clear

relationship between the degree of calcification and the carbonate chemistry of sea water are established yet, but rather different correlations seem to exist in different areas of the world oceans and under different oceanographic conditions, so the question remains open.
MSP: This point has somehow been made also by reviewer #1 and it has already been addressed, highlighting the temperature as a controlling factor.

R#2: Page 11 line 12: also Malinverno et al., 2016 show the shift in dominance from coccolithophores to diatoms in water samples across the PF / sACCf, so this could be cited.
MSP: We thank reviewer#2 for suggesting this paper. We included this reference and reworded the sentence as follows: "In any case the southernmost extent of coccolithophores is also influenced by the clear dominance of diatoms south of the PF, as suggested by the high diatom concentration (valves/g dry sediment) and biogenic opal content recorded in surface sediment samples from the AZ of the Drake Passage (Cárdenas et al., 2018) and from Pacific Southern Ocean extant plankton studies (e.g., Saavedra-Pellitero et al., 2014; Malinverno et al., 2016)."

R#2: Typos: Page 1, line 22: classified -> identified
MSP: We changed it to "identified".

R#2: Page 2, line 6: delete "substantial" which is repeated twice
MSP: The extra "substantial" was deleted, as suggested also by reviewer#1.

R#2: Page 2, line 7: dissolved carbon » dissolved inorganic carbon
MSP: We changed it to "dissolved inorganic carbon".

R#2: Page2, line 16: phosphate is mis-spelled
MSP: We corrected it.

R#2: Page 2, line 28: the future -> in the future
MSP: We changed it.

R#2: Page 4,line 6: (2004) is repeated
MSP: We deleted one of the "(2004)".

R#2: Page 7 line 20: this taxa -> this taxon
We changed it to "taxon".

R#2: Page 8 line 1: later -> latter
MSP: We corrected it.

R#2: Page 13 line 5: established -> established by
MSP: We added "by".
* * *
**Additional changes:**

Title: Calcification and latitudinal distribution of extant coccolithophores across the Drake Passage during late austral summer 2016
We would like to add the word "latitudinal" to the title, because it makes it more precise.

Page 1, L8: includes…a unicellular
Page 1, L14. To avoid repetition we changed "expected" to "predicted"
Page 1, L27: succession
Page 3, L9: *E. huxleyi* (because the whole name was previously mentioned)
Page 3, L31: a instead of an
Page 4, L14: missing space
Page 4, L33: we deleted the dash in "derived- variables"
Page 5, L2: The comma should not superscript
Page 6, L5: We added "H=" to equation (1)
Page 6, L27: sp.
Page 8, L20: we deleted an extra space
Page 8, L26: *Papposphaera* sp.
Page 9, L18: environment
Page 9, L22: we changed "maximum" to "maximal"
Page 10, L3: pg instead of picrograms
Page 10, L10: portray
Page 10, L26: occurring….Antarctic Circumpolar Current
Page 11, L31: mesoscale
Page 12: We changed the title of section 4.2, because it seemed random/preliminary title.
Page 12, L 22: has already been
Page 13, L7: might instead of may
Page 13, L13: spp.
Page 13, L31: sp.,
Page 13, L31: lightly calcified instead of lightly-calcified
Page 14, L10: observed
Page 14, L10: estimated masses instead of estimated mass values
Page 14, L27: succession
Page 15, L21: sp.
Page 16: We changed the title to just "Conclusions"
Page 16: We added the link for the Pangaea data repository:
https://doi.org/10.1594/PANGAEA.901294.
References: We added the missing reference for Schlitzer (2015)
Fig. 13 caption: we deleted some extra commas

Additionally we added/deleted some missing spaces/commas throughout the manuscript.
In the conclusions we capitalized some of the terms (e.g., Polar Front), just for consistency.

[revised manuscript text omitted]